# Uncertainty-aware Preference Alignment for Diffusion Policies

## Abstract

Recent advancements in diffusion policies have demonstrated promising performance in decision-making tasks. To align these policies with human preferences, a common approach is incorporating Preference-based Reinforcement Learning (PbRL) into policy tuning. However, since preference data is practically collected from populations with different backgrounds, a key challenge lies in handling the inherent uncertainties in people's preferences during policy updates. To address this challenge, we propose the Diff-UAPA algorithm, designed for uncertainty-aware preference alignment in diffusion policies. Specifically, Diff-UAPA introduces a novel iterative preference alignment framework in which the diffusion policy adapts incrementally to preferences from different user groups. To accommodate this online learning paradigm, Diff-UAPA employs a maximum posterior objective, which aligns the diffusion policy with regret-based preferences under the guidance of an informative Beta prior. This approach enables direct optimization of the diffusion policy without specifying any reward functions, while effectively mitigating the influence of inconsistent preferences across different user groups. We conduct extensive experiments across various robot control tasks and diverse human preference configurations, demonstrating the robustness and reliability of Diff-UAPA in achieving effective preference alignment.

## 1. Introduction

Reinforcement Learning (RL) algorithms commonly employ either deterministic or Gaussian policies to tackle sequential decision-making tasks by optimizing cumulative rewards (Sutton & Barto, 2018; Wang et al., 2022). Although these RL policies have demonstrated notable success across a wide range of applications (Mnih et al., 2015; Silver et al., 2016; Fang et al., 2019), they may struggle with learning multi-modal policies, which may hinder their ability to generalize effectively and lead to suboptimal performance in complex environments (Zhu et al., 2023). Recently, diffusion models have gained attention due to their strong modeling capabilities (Ho et al., 2020; Song et al., 2020). As a result, more studies have investigated the application of diffusion models in RL tasks, particularly in leveraging diffusion models as policies to model complex action distributions and behaviors (Wang et al., 2023; Chen et al., 2023a; Kang et al., 2023a; Lu et al., 2023; Chi et al., 2023). To learn a diffusion policy that generates desired outputs, recent approaches have leveraged Preference-based Reinforcement Learning (PbRL) (Christiano et al., 2017) techniques, which address a learning-to-rank problem using preference data, enabling alignment with human intentions (Wallace et al., 2024; Dong et al., 2024; Shan et al., 2024).

In practice, preferences are typically gathered from a diverse population, encompassing a wide range of expertise, perspectives, and beliefs. This diversity presents a significant challenge, as preferences from different user groups may conflict or evolve over time, introducing great uncertainties during policy updates. To ensure more reliable preference alignment, this necessitates the development of a policy that could account for the uncertainty arising from potentially inconsistent preferences. However, common PbRL approaches are typically based on the Bradley-Terry model (Bradley & Terry, 1952) with maximum likelihood estimation, which lacks sensitivity to the inherent uncertainties from preference datasets.

To address the uncertainties in preference alignment, several methods (Liang et al., 2022; Shin et al., 2023; Xue et al., 2024) have employed techniques such as ensemble models and Bayesian dropout. However, the underlying mechanism by which the estimated ensembles correlate with uncertainty remains largely unexplained. Motivated by recent work (Xu et al., 2025), which proposes learning a distributional reward model using a Maximum A Posteriori (MAP) objective to address epistemic uncertainty from an offline preference dataset, we explore how to bypass the reward learning and develop an uncertainty-aware algorithm beyond the offline setting for aligning diffusion policies.

[1]Anonymous Institution, Anonymous City, Anonymous Region, Anonymous Country. Correspondence to: Anonymous Author <anon.email@domain.com>.

Preliminary work. Under review by the International Conference on Machine Learning (ICML). Do not distribute.

*Figure 1.* The framework of Diff-UAPA. Given the potentially inconsistent preference dataset ranked by diverse humans, we first learn a Beta prior to capture uncertainties, and then derive a Maximum A Posteriori (MAP) objective to align the diffusion policies.

In this work, we introduce Uncertainty-aware Preference Alignment for Diffusion Policies (Diff-UAPA),a novel algorithm designed to align diffusion policies with human preferences using an uncertainty-aware objective, as illustrated in Figure 1. Specifically, we introduce an iterative preference alignment framework, in which the diffusion policy progressively adapts to the labels coming from different user groups, each of which may have distinct preferences. To address this challenge, Diff-UAPA involves learning an informative Beta prior that captures the uncertainty arising from diverse human preferences. By interpreting preference alignment as a voting process, we demonstrate that the Beta distribution is sensitive to the uncertainty among compared trajectories, assigning high confidence to trajectories in which the majority of human raters share a common preference and low confidence to those with divergent preferences. To ensure computational tractability, we parameterize the Beta distribution with neural networks and train the model via variational inference.

Guided by the informative Beta prior, Diff-UAPA aligns the diffusion policy with a regret-based preference model, which inherently defines a unified Maximum A Posteriori (MAP) objective. This method enables direct optimization of the diffusion policy without requiring a reward function, while also effectively accounting for the uncertainties arising from noisy preferences across diverse user groups.

To evaluate the empirical performance of Diff-UAPA, we conduct extensive experiments across a diverse range of robot manipulation and locomotion tasks, comparing its performance against recently proposed baseline methods. Furthermore, we investigate its effectiveness using heterogeneous human preference data, including synthesized, realistic, and noisy preferences. The results demonstrate the robustness and reliability of Diff-UAPA in handling varying levels of uncertainty in preference data.

## 2. Related Works

### 2.1. Preference-based Reinforcement Learning

Preference-based Reinforcement Learning (PbRL) is a pivotal approach for aligning agents with human intent, particularly in scenarios where specifying explicit reward functions is challenging (MacGlashan et al., 2017; Warnell et al.,

2018; Wirth et al., 2017). Previous works generally adopt a two-step procedure, where an explicit reward model is first inferred from human preferences using the Bradley-Terry model (Bradley & Terry, 1952), followed by training an RL agent to optimize the learned reward (Christiano et al., 2017; Ibarz et al., 2018). Building on this framework, several methods (Lee et al., 2021; Park et al., 2022; Hejna III & Sadigh, 2023; Liu et al., 2022; Liang et al., 2022; Hwang et al., 2023; Choi et al., 2024) have enhanced the learning process, focusing on improving efficiency and capability. In terms of preference modeling, while earlier works generally assume that preferences are generated based on the sum of Markovian rewards, recent studies (Kim et al., 2023; Verma & Metcalf, 2024) have proposed modeling preferences using non-Markovian rewards. Instead of learning an explicit reward model, another line of research focuses on directly optimizing policies or extracting value functions from human preferences (An et al., 2023; Hejna et al., 2024; Hejna & Sadigh, 2024). This approach is more straightforward, avoiding the biases and information bottleneck from intermediate reward modeling (Kang et al., 2023b).

### 2.2. Diffusion Policy for Decision Making

Diffusion models have outperformed earlier generative models in both sample quality and training stability, gaining significant attention across various domains, including offline RL (Janner et al., 2022; Ajay et al., 2023), online RL (Yang et al., 2023; Chen et al., 2024), and robotics (Sridhar et al., 2024; Chen et al., 2023b; Xu et al., 2023). Recent advancements have leveraged diffusion models as RL policies to capture arbitrary action distributions, improving decision-making capabilities (Zhu et al., 2023). Among these works, Diffusion-QL (Wang et al., 2023), first integrated diffusion policies into the Q-learning framework. Following this, SfBC (Chen et al., 2023a) refined policy learning by decoupling behavior learning from action evaluation, while CEP (Lu et al., 2023) extended this framework to enable sampling from broader energy-guided distributions. CPQL (Chen et al., 2024) introduced consistency models to accelerate training and sampling, and EQP (Kang et al., 2023a) enhanced training efficiency with single-step model predictions for action approximations. In preference-based tasks, AlignDiff (Dong et al., 2024) utilized diffusion

planners to generate trajectories aligned with human preferences through a two-step procedure, while FKPD (Shan et al., 2024) introduced a one-step framework for direct alignment. However, these methods often fail to account for the uncertainties inherent in human preferences. How to handle these uncertainties when aligning diffusion policies remains a critical challenge (Casper et al., 2023).

## 3. Problem Formulation

**Preference-based Reinforcement Learning (PbRL).** Reinforcement Learning (RL) algorithms (Sutton & Barto, 2018) typically consider an episodic Markov Decision Process (MDP), which is formally defined as a tuple $\mathcal{M} = (\mathcal{S}, \mathcal{A}, p_{\mathcal{R}}, p_{\mathcal{T}}, \gamma, T, \mu_0)$, where: 1) $\mathcal{S}$ and $\mathcal{A}$ represent the state and action spaces, 2) $p_{\mathcal{R}}(r|s, a)$ and $p_{\mathcal{T}}(s'|s, a)$ define the (stochastic) reward and transition functions, 3) $\gamma \in (0, 1]$ is the discount factor, 4) $\mu_0$ denotes the initial state distribution and 5) $T \in (0, \infty)$ denotes a non-fixed planning horizon, and the games is reset when the agent reaches a terminating or goal state at a time step $T$. In many applications, the reward function is not directly available, reducing the episodic MDP to a reward-free MDP $\mathcal{M}_{/r}$. To resolve this challenge, PbRL algorithms (Christiano et al., 2017) proposed learning the reward function from human preferences datatset. Specifically, given an unlabeled dataset of trajectory segments $\mathcal{D}_\tau = \{\tau\}$, humans randomly select a pair of trajectories and rank them according to their preferences on the optimality. By recording these pair-wise comparisons, we create a preference dataset $\mathcal{D}_{\text{pref}} = \{(\tau^w, \tau^l)\}$, where each trajectory segment of length $k$ is defined as $\tau = (s_1, a_1, s_2, a_2, \ldots, s_k, a_k)$, and $\tau^w$ is preferred over $\tau^l$. Based on this dataset, recent methods (Christiano et al., 2017; Ibarz et al., 2018) commonly infer the rewards by employing the Bradley-Terry model (Bradley & Terry, 1952) with maximum likelihood estimation (MLE).

**Uncertainty Model in Preference Alignment.** The Bradley-Terry model (Bradley & Terry, 1952) can effectively model pairwise comparisons, whether by explicitly inferring a reward function (Christiano et al., 2017; Lee et al., 2021; Park et al., 2022) or by directly aligning policies with preferences (Hejna et al., 2024; An et al., 2023). However, this approach fails to account for the inherent uncertainty in human preferences (Newman, 2023; Xu et al., 2025), particularly when these preferences are collected from a diverse population with varying levels of expertise, perspectives, and beliefs. More critically, for continuous learning, the policy must adapt dynamically to preferences from different user groups, which often arrive incrementally over time. To resolve these challenges, we study an iterative preference alignment problem:

**Definition 3.1.** (Iterative Preference Alignment) Let $\mathcal{D}_\tau = \tau$ denote the trajectory dataset, and let $\mathcal{D}_{\text{pair}}^n = (\tau^i, \tau^j)$

represent the pairwise comparisons dataset constructed at the $n^{th}$ iteration. These comparisons are generated by 1) sampling pairs of trajectories from $\mathcal{D}\tau$ and 2) inviting a group of annotators to label them. The algorithm must progressively align the policy $\pi$ with the preference dataset $\mathcal{D}_{\text{pair}}^n$ at each round $n \in [1, N]$ in an online manner.

In this setting, different groups of human annotators may provide inconsistent or even conflicting preferences for the same pair of trajectories (Liang et al., 2022; Shin et al., 2023; Xue et al., 2024). The problem solver must dynamically adapt the policy to iteratively updated preference signals while ensuring that the learned policy effectively represents general human preferences by performing online updates.

Additionally, apart from the preference signals, the trajectory dataset $\mathcal{D}_\tau$ can in principle be updated based on interaction from the environment. However, in practice, such interactions are not always available, and thus we assume $\mathcal{D}_\tau$ mainly records only offline trajectories. The primary challenge is to stabilize the policy optimization process and learn a reliable control policy by effectively managing the aleatoric uncertainty inherent in stochastic and potentially inconsistent preference signals on the provided trajectories.

**Preference Alignment for Diffusion Policies.** While previous PbRL methods have commonly focused on policies modeled by feed-forward neural networks, recent studies highlight the superior control performance achieved by diffusion-based policies (Zhu et al., 2023). Denoising diffusion models (Ho et al., 2020) represent a class of generative models characterized by an iterative diffusion and denoising process. Diffusion models have gained significant attention in decision-making tasks due to their ability to represent complex multi-modal distributions (Zhu et al., 2023). This capability is crucial for characterizing the policy function $\pi_\theta(a|s)$, surpassing previous deterministic or Gaussian-based policies (Chi et al., 2023; Wang et al., 2023). Diffusion policies are typically formulated as conditional generative models as follows[1]:

$$\pi_\theta(a_t|s_t) = \int \mathcal{N}(a_t^I; \mathbf{0}, \mathbf{I}) \prod_{i=1}^{I} \pi_\theta(a_t^{i-1}|a_t^i, s_t) da_t^{1:I}, \tag{1}$$

where $\pi_\theta(a_t^{i-1}|a_t^i, s_t)$ is often parameterized as Gaussian with fixed timestep-dependent covariances as $\mathcal{N}(a_t^{i-1}|\mu_\theta(a_t^i, s_t, i), \Sigma^i)$. Although diffusion policies can be trained from offline datasets, their performance is often constrained by the size, quality, and availability of the expert demonstration dataset. As a result, many previous methods have utilized RL algorithms to improve these policies with experience data sampled from an interactive MDP

---

[1]In this work, we use superscripts ($i \in \{0, 1, \ldots, I\}$ to denote diffusion timesteps and subscripts ($t \in \{0, 1, \ldots, T\}$) to denote trajectory timesteps.

environment (Kang et al., 2023a; Psenka et al., 2024). In this setting, recent research (Wallace et al., 2024) proposed leveraging Direct Preference Optimization (DPO) (Rafailov et al., 2023) to align diffusion policies with human preferences based on $\mathcal{D}_{\text{pref}}$. Specifically, DPO algorithms directly optimize policies without learning a reward model, thereby significantly enhancing the efficiency and stability of the training process. To train $\pi_\theta$, the maximum likelihood objective for state-action pairs is defined as follows:

$$L(\theta) = -\mathbb{E}\Big[ \log \sigma \Big( -\lambda I \cdot \tag{2}$$

$$\Big( (\|\epsilon^w - \epsilon_\theta(a^{i,w}, s^w, i)\|_2^2 - \|\epsilon^w - \epsilon_{\text{ref}}(a^{i,w}, s^w, i)\|_2^2)$$

$$- (\|\epsilon^l - \epsilon_\theta(a^{i,l}, s^l, i)\|_2^2 - \|\epsilon^l - \epsilon_{\text{ref}}(a^{i,l}, s^l, i)\|_2^2) \Big) \Big],$$

where 1) $\big( (s^w, a^{0,w}), (s^l, a^{0,l}) \big) \sim \mathcal{D}_{\text{pref}}$ are state-action samples from preference dataset, 2) $i \sim \mathcal{U}(0, I)$ is the diffusion timestep, and 3) $a^{i,w/l} \sim q(a^{i,w/l}|a^{0,w/l}, s^{w/l})$ denotes the action $a^{0,w/l}$ corrupted with noise $\epsilon^{w/l}$ after $i$ diffusion steps, as defined in (Ho et al., 2020). In this study, we explore addressing the iterative preference alignment problem by aligning human preferences with a diffusion policy model.

## 4. Uncertainty-Aware Preference Alignment for Diffusion Policies

In this section, we outline our approach for aligning a diffusion policy with human preferences while effectively accounting for uncertainty. Specifically, we present: 1) a Maximum Likelihood Estimation (MLE) objective for diffusion policy alignment, based on maximum entropy framework and direct preference optimization (Section 4.1), 2) a Maximum A Posteriori (MAP) objective that incorporates a Beta prior model for capturing the underlying uncertainties (Section 4.2), and 3) the training procedure for the Beta prior model (Section 4.3).

### 4.1. Maximum Likelihood Diffusion Policy Alignment

**MaxEnt Alignment under Regret Preference.** Following previous works on preference alignment (Hejna et al., 2024; Rafailov et al., 2024; Ouyang et al., 2022), we adopt the Maximum Entropy (MaxEnt) RL framework. In this approach, the objective is to learn a policy $\pi_\theta$ that not only maximizes its cumulative discounted rewards but also incorporates the causal entropy, while regularizing the KL-divergence from a reference policy (Ziebart, 2010):

$$\max_\pi \mathbb{E}_\pi \left[ \sum_{t=0}^{T} \gamma^t (r(s_t, a_t) - \alpha \log \frac{\pi(a_t|s_t)}{\pi_{\text{ref}}(a_t|s_t)}) \right], \tag{3}$$

Here, $\alpha$ determines the weight of entropy in the optimization objective. Upon learning an optimal policy $\pi^*$, we can compute the corresponding optimal state-value function $V^*(s_t)$, the optimal state-action value function $Q^*(s_t, a_t)$, and the

optimal advantage function $A^*(s_t, a_t) \triangleq Q^*(s_t, a_t) - V^*(s_t)$. More importantly, in the MaxEnt RL setting, the optimal advantage function is proportional to the log-likelihood of the optimal and reference policy (Haarnoja et al., 2017; Hejna et al., 2024):

$$A^*(s_t, a_t) = \alpha \log \frac{\pi^*(a_t|s_t)}{\pi_{\text{ref}}(a_t|s_t)}. \tag{4}$$

To stabilize the process of preference alignment, we follow (Knox et al., 2022) and base the preference alignment on discounted regrets, defined as $-\sum \gamma^t (V^(s_t) - Q^(s_t, a_t))$. In this framework, a trajectory segment is preferred if it incurs lower regret compared to the intended optimal policy, so that the preference between trajectory segments $(\tau^w, \tau^l)$ can be modeled as:

$$P_{A^*}(\tau^w \succ \tau^l) = \tag{5}$$

$$\frac{\exp \sum_{t=0}^{k} \gamma^t A^*(s_t^w, a_t^w)}{\exp \sum_{t=0}^{k} \gamma^t A^*(s_t^w, a_t^w) + \exp \sum_{t=0}^{k} \gamma^t A^*(s_t^l, a_t^l)}.$$

By substituting Equation (4) into Equation (5), the advantage function $A^*$ can be replaced by the optimal policy $\pi^*$ under the MaxEnt framework. The learned policy $\pi_\theta$ can then be optimized through maximum the likelihood of generating preferences as follows (Hejna et al., 2024):

$$\mathcal{L}_{\text{CPL}}^{(\tau^w, \tau^l)}(\theta) = -\log \sigma \Big( \alpha \cdot \tag{6}$$

$$\big( \sum_{t=0}^{k} \gamma^t \log \frac{\pi_\theta(a_t^w|s_t^w)}{\pi_{\text{ref}}(a_t^w|s_t^w)} - \sum_{t=0}^{k} \gamma^t \log \frac{\pi_\theta(a_t^l|s_t^l)}{\pi_{\text{ref}}(a_t^l|s_t^l)} \big) \Big),$$

**Diffusion Policy Alignment.** To adapt the previous model to aligning the diffusion policy $\pi_\theta(a_t|s_t)$ as defined in Equation (1), a primary difficulty is due to the intractability of diffusion policy $\pi_\theta(a_t|s_t) = \int \pi_\theta(a_t^{0:I}|s_t) \mathrm{d}a_t^{1:I}$, as it requires marginalizing over all possible diffusion paths $(a_t^1, a_t^2, \ldots, a_t^I)$ that lead to $a_t^0$. To address it, we propose modeling the chain reward function (Wallace et al., 2024):

$$r(s_t, a_t^0) = \mathbb{E}_{\pi_\theta(a_t^{1:I}|a_t^0, s_t)}[r(s_t, a_t^{0:I})]. \tag{7}$$

The optimal chain advantage function can be defined as:

$$A^*(s_t, a_t^0) = \mathbb{E}_{\pi_\theta^*(a_t^{1:I}|a_t^0, s_t)} \left[ A^*(s_t, a_t^{0:I}) \right] \tag{8}$$

$$= \mathbb{E}_{\pi_\theta^*(a_t^{1:I}|a_t^0, s_t)} \left[ \alpha \log \frac{\pi_\theta^*(a_t^{0:I}|s_t)}{\pi_{\text{ref}}(a_t^{0:I}|s_t)} \right]. \tag{9}$$

In principle, we can interpret the latent diffusion actions as a unified chain action $\overline{a_t} = a_t^{0:I}$, despite the final output being determined by $a_t^0$. This perspective allows us to reformulate Equation (3) in terms of the diffusion policy:

$$\max_{\pi_\theta} \mathbb{E}_{\pi_\theta(\overline{a_t}|s_t)} \left[ \sum_{t=0}^{T} \gamma^t (r(s_t, \overline{a_t}) - \alpha \log \frac{\pi_\theta(\overline{a_t}|s_t)}{\pi_{\text{ref}}(\overline{a_t}|s_t)}) \right]. \tag{10}$$

This objective is defined over the entire diffusion path $\overline{a_t}$, which aims to maximize the cumulative rewards and the entropy within a trajectory across the reverse process.

By paralleling from Equation (3) to Equation (6), the objective in (10) can be directly optimized with respect to the diffusion policy $\pi_\theta(\overline{a_t}|s_t)$ by maximizing the following likelihood:

$$\mathcal{L}_{1,\text{MLE}}^{(\tau^w,\tau^l)}(\theta) = -\log\sigma\Big(\alpha\cdot \tag{11}$$

$$\Big(\sum_{t=0}^{k}\mathbb{E}_{\pi_\theta(a_t^{1:I,w}|s_t^w,a_t^{0,w})}\left[\gamma^t\log\frac{\pi_\theta(\overline{a_t^w}|s_t^w)}{\pi_{\text{ref}}(\overline{a_t^w}|s_t^w)}\right]$$

$$-\sum_{t=0}^{k}\mathbb{E}_{\pi_\theta(a_t^{1:I,l}|s_t^l,a_t^{0,l})}\left[\gamma^t\log\frac{\pi_\theta(\overline{a_t^l}|s_t^l)}{\pi_{\text{ref}}(\overline{a_t^l}|s_t^l)}\right]\Big)\Big),$$

where $\sigma$ is the sigmoid function. However, major challenges in optimizing this objective lie in: 1) *inefficiency*, due to the sequential computation required across many timesteps, and 2) *intractability*, stemming from the need to evaluate the joint distribution. Inspired by Wallace et al. (2024), we leverage Jensen's inequality and the convexity of the $-\log\sigma$ function to move the expectation operator outside, thereby improving efficiency. Additionally, we approximate the reverse process $\pi_\theta(a_t^{1:I}|s_t)$ using the forward process $q(a_t^{1:I}|s_t)$, which makes the problem more tractable. With some algebra, we derive the following loss function:

$$\mathcal{L}_{1,\text{MLE}}^{(\tau^w,\tau^l)}(\theta) \leq -\mathbb{E}_{\substack{a_t^{i,w}\sim q(a_t^{i,w}|a_t^{0,w},s_t^w),\\ a_t^{i,l}\sim q(a_t^{i,l}|a_t^{0,l},s_t^l)}}\Big[\log\sigma\Big(-\alpha I\cdot$$

$$\Big(\sum_{t=0}^{k}\gamma^t(\|\epsilon^w - \epsilon_\theta(a_t^{i,w},s_t^w,i)\|_2^2 - \|\epsilon^w - \epsilon_{\text{ref}}(a_t^{n,w},s_t^w,i)\|_2^2)$$

$$-\sum_{t=0}^{k}\gamma^t(\|\epsilon^l - \epsilon_\theta(a_t^{i,l},s_t^l,i)\|_2^2 - \|\epsilon^l - \epsilon_{\text{ref}}(a_t^{i,l},s_t^l,i)\|^2)\Big)\Big)\Big]$$

$$= \mathcal{L}_{2,\text{MLE}}^{(\tau^w,\tau^l)}(\theta), \tag{12}$$

The detailed deviation is shown in Appendix A.

### 4.2. Bayesian Alignment with Informative Beta Prior

The regret preference model (Equation (5)) represents the likelihood of generating human preferences based on the advantage function. The corresponding maximum likelihood objective implicitly assumes a uniform prior over $\sum_{t=0}^{k}\gamma^t A^*(s_t,a_t)$, which does not account for the uncertainty within the preference dataset, and may lead to divergence in the parameters of the learned policy (Newman, 2023; Xu et al., 2025). We present how to derive a more informative prior as follows.

Since human feedback is based on two trajectories rather than individual state-action pairs, we assume that the strength of a trajectory is defined by its trajectory-level advantage, represented by its discounted cumulative advan-

tages under the diffusion policy $\pi_\theta$:

$$A^{\pi_\theta}(\tau) = \sum_{t=0}^{k}\gamma^t A^{\pi_\theta}(s_t,a_t)$$

$$= \sum_{t=0}^{k}\gamma^t \mathbb{E}_{\pi_\theta(a_t^{1:I}|a_t^0,s_t)}\left[A^{\pi_\theta}(s_t,\overline{a_t})\right]. \tag{13}$$

The average strength of the trajectories under policy $\pi_\theta$ is then defined as:

$$\bar{A}^{\pi_\theta} = \mathbb{E}_{\tau\sim\mathcal{D}_\tau}A_\theta(\tau) = \frac{1}{|\mathcal{D}_{\text{pref}}|}\sum_{\tau\in\mathcal{D}_{\text{pref}}}A^{\pi_\theta}(\tau). \tag{14}$$

Therefore, the probability of a trajectory with strength $A^{\pi_\theta}(\tau)$ winning against the average candidate is $\phi(\tau) = \sigma(A^{\pi_\theta}(\tau) - \bar{A}^{\pi_\theta}) \in (0,1)$. By applying the chain rule, the prior on the advantage function can be defined as:

$$p_0(A^{\pi_\theta}(\tau)) = p_0(\phi(\tau))\frac{d\phi(\tau)}{dA^{\pi_\theta}(\tau)}$$

$$= p_0(\phi(\tau))\sigma'(A^{\pi_\theta}(\tau) - \bar{A}^{\pi_\theta})(1 - \frac{1}{|\mathcal{D}_{\text{pref}}|}). \tag{15}$$

This prior reflects our initial belief about the strength of different trajectories within the dataset. Motivated by Xu et al. (2025), we use the Beta distribution as the informative prior, i.e., $p_0(\phi(\tau)) = \text{Beta}(\phi(\tau);\alpha,\beta)$. The main benefits of the Beta distribution are: 1) it is the conjugate prior for the Bernoulli distribution, and $\phi(\tau)$ naturally ranges from $(0,1)$, which simplifies updates with new evidence, and 2) the parameters $\alpha$ and $\beta$ can intuitively represent the counts of *preferred* and *unpreferred* human feedback. By reformulating Eq. (15), we present the following proposition:

**Proposition 4.1.** *Let the informative prior $p_0(\phi(\tau))$ be a Beta distribution $Beta(\phi(\tau);\alpha,\beta)$. This prior can effectively capture the uncertainty arising from the iterative preference alignment process (Definition 3.1). Consequently, the prior on the strength of a trajectory is proportional to $Beta((\phi(\tau);\alpha+1,\beta+1))$, i.e., $p_0(A^{\pi_\theta}(\tau)) \propto Beta(\phi(\tau);\alpha+1,\beta+1)$.*

The proof is shown in Appendix C. The corresponding prior loss can then be derived in a manner similar to the derivation of the maximum likelihood loss (Eq. 11):

$$\mathcal{L}_{1,\text{prior}}^{\tau}(\theta)$$

$$= -\log\text{Beta}(\phi(\tau);\alpha+1,\beta+1)$$

$$\leq -\mathbb{E}\Big[\log\text{Beta}\Big(\sigma\Big(-\alpha I\cdot\Big(\sum_{t=0}^{k}\gamma^t(\big\|\epsilon - \epsilon_\theta(a_t^i,s_t,i)\big\|_2^2 -$$

$$\big\|\epsilon - \epsilon_{\text{ref}}(a_t^i,s_t,i)\big\|_2^2) - \sum_{\tau\in\mathcal{D}_{\text{pref}},t=0}^{k}\frac{\gamma^t}{|\mathcal{D}_{\text{pref}}|}(\big\|\epsilon - \epsilon_\theta(a_t^i,s_t,i)\big\|^2 -$$

$$\big\|\epsilon - \epsilon_{\text{ref}}(a_t^i,s_t,i)\big\|^2)\Big)\Big);\alpha+1,\beta+1\Big)\Big]$$

$$= \mathcal{L}_{2,\text{prior}}^{\tau}(\pi_\theta) \tag{16}$$

Appendix B shows the detailed proof. Equation (16) can be interpreted as guiding the policy to align the estimated advantage function for trajectories with their prior distribution. Since $P_{\text{MAP}}(A(\tau)) \propto p_0(A(\tau)) \cdot P_{\text{MLE}}(A(\tau))$, by incorporating the prior into the MLE objective and maximizing the log form of the posterior, we can derive the Diff-UAPA loss:

$$\mathcal{L}_{\text{Diff-UAPA}}(\theta) = \mathbb{E}_{(\tau^w, \tau^l) \sim \mathcal{D}_{\text{pref}}} \Big[ \mathcal{L}_{2,\text{MLE}}^{(\tau^w, \tau^l)}(\pi_\theta)$$
$$+ \mathcal{L}_{2,\text{prior}}^{\tau^w}(\pi_\theta) + \mathcal{L}_{2,\text{prior}}^{\tau^l}(\pi_\theta) \Big]. \quad (17)$$

Maximizing the posterior probability, rather than the likelihood, incorporates prior knowledge and regularizes advantage values, preventing divergence. We introduce how to estimate the Beta prior in the following section.

## 4.3. Training the Beta Prior Model

To learn the Beta prior $p_0(\phi(\tau)|\mathcal{D}_{\text{pref}}) = \text{Beta}(\phi(\tau); \alpha, \beta)$ in continuous spaces, following (Xu et al., 2025), we propose using a variational inference approach to approximate it by estimating the approximate posterior $q_\xi(\phi(\tau)|\mathcal{D}_{\text{pref}})$, i.e., $p_0(\phi(\tau)|\mathcal{D}_{\text{pref}}) \simeq q_\xi(\phi(\tau)|\mathcal{D}_{\text{pref}})$, where $\xi$ is the model parameters. The objective is to minimize the Kullback-Leibler (KL) divergence between the prior and posterior, which is equivalent to maximizing the Evidence Lower Bound (ELBO). This leads to the following interpretation of the corresponding trajectory-wise objective (Xu et al., 2025):

$$\max_\xi \mathbb{E}_\tau \Big[ \mathbb{E}_{q_\xi, (\tau^w, \tau^l) \in \mathcal{D}_{\text{pref}}} [\log \phi(\tau^w)] - \quad (18)$$

$$\mathbb{E}_{q_\xi, (\tau^w, \tau^l) \in \mathcal{D}_{\text{pref}}} [\log \phi(\tau^l)] - D_{\text{KL}} [q_\xi(\phi(\tau)|\tau) \parallel p(\phi(\tau))] \Big],$$

where 1) $q_\xi(\phi(\tau)|\tau) = \text{Beta}(\alpha_\tau, \beta_\tau)$, where $[\alpha_\tau, \beta_\tau] = f_\xi^{\text{Beta}}(\tau)$ and $f_\xi^{\text{Beta}}$ denotes a neural network, 2) $p(\phi(\tau)) = \text{Beta}(\alpha_0, \beta_0)$, with $\alpha_0, \beta_0$ specifying our prior belief (we set $\alpha_0 = \beta_0 = 1$ in this work), and 3) $\phi(\tau)$ represents the Bernoulli probability that $\tau^w$ is ranked higher than $\tau^l$. The first two terms aim to optimize the parameter $\xi$ to align with the preference dataset, while the final KL-divergence term ensures the posterior distribution does not deviate too far from the prior belief, which can be optimized using the Dirichlet VAE approach (Joo et al., 2020).

In this work, we implement $f_\xi^{\text{Beta}}(\tau)$ using a transformer-based neural network (Vaswani, 2017), where the trajectory $\tau$ is fed as input and $[\alpha_\tau, \beta_\tau]$ is produced as the output to form the Beta prior distribution. The complete Diff-UAPA algorithm is shown in Algorithm 1.

## 5. Empirical Evaluation

In this section, we empirically evaluate the proposed Diff-UAPA algorithm on four robot manipulation tasks across two environments (Section 5.1) and locomotion tasks with

---

**Algorithm 1** Uncertainty-aware Preference Alignment for Diffusion Policies (Diff-UAPA)

---
1: **Input:** Trajectory dataset $\mathcal{D}_\tau$, preference dataset $\mathcal{D}_{\text{pref}}$, prior training epochs $M$, policy training epochs $N$.
2: Initialize Beta prior model $f_\xi^{\text{Beta}}(\tau)$, reference policy $\pi_{\text{ref}}(a|s)$, and diffusion policy $\pi_\theta(a|s)$.
3: Learn $\pi_{\text{ref}}$ based on $\mathcal{D}_\tau$ through behavior cloning.
4: **for** $m = 1, \cdots, M$ **do**
5:     Update the Beta prior $f_\xi^{\text{Beta}}$ with objective (18).
6: **end for**
7: **for** $n = 1, \cdots, N$ **do**
8:     Update the diffusion policy $\pi_\theta$ by minimizing Eq. (17).
9: **end for**

---

real human preferences (Section 5.2), where preferences are continuously updated and may exhibit inconsistencies. Additionally, we evaluate the noise sensitivity of the proposed method under different levels of preference inconsistency (Section 5.3).

**Experiment Settings.** We evaluate the methods on three tasks in Robomimic (Mandlekar et al., 2021) and one long-horizon Franka Kitchen (Gupta et al., 2019) environment for manipulation tasks, as well as two environments in D4RL (Fu et al., 2020) with real human preferences for locomotion tasks. Our experiments consist of four rounds of iterative updates, with each round consisting of a fixed number of training episodes. To account for potential inconsistencies in human preferences, we introduce a reverse rate into the ground-truth preference data. Specifically, in each update round, we randomly select 20% of trajectory pairs and apply a 50% reversal rate by swapping the winner and the loser. The learning rate is reset at the beginning of each round to enhance stability and convergence. After training, the policy is evaluated over 10 episodes in 56 parallel environments. Each experiment is repeated using three different random seeds, and the mean $\pm$ standard deviation (std) of the results is reported. More experimental details can be found in Appendix D.1.

**Comparison Methods.** We utilize two baseline policies: the Gaussian-based policy from Behavior Transformer (**BET**) (Shafiullah et al., 2022) and the Diffusion Policy (**Diff**) (Chi et al., 2023). In BET, we apply focal loss (Mukhoti et al., 2020) for preference-based learning and leverage the full set of trajectories in the preference dataset for training the diffusion policy.

Building on BET, we propose the following comparison methods: 1) BET-Direct Preference Optimization (**BET-DPO**) and 2) BET-Contrastive Preference Learning (**BET-CPL**), which leverage direct preference optimization

*Table 1.* Success rates (in percentage) of all methods across the Robomimic and Kitchen tasks, with each value presented as the mean $\pm$ std, computed over 3 training seeds and 560 evaluation episodes. The best results for each task are highlighted in bold. For the Kitchen task, p$x$ indicates the frequency of interaction with $x$ or more objects.

| | Robomimic | | | Kitchen | | | |
|---|---|---|---|---|---|---|---|
| |  | | |  | | | |
| | Lift | Can | Square | p1 | p2 | p3 | p4 |
| BET | $43.6 \pm 3.8$ | $48.8 \pm 3.1$ | $55.1 \pm 2.0$ | $96.4 \pm 1.2$ | $96.2 \pm 1.0$ | $76.6 \pm 1.3$ | $44.6 \pm 2.0$ |
| BET-CPL | $49.2 \pm 4.4$ | $42.1 \pm 1.1$ | $57.6 \pm 2.3$ | $97.0 \pm 1.0$ | $96.4 \pm 0.5$ | $88.4 \pm 2.3$ | $62.6 \pm 2.0$ |
| BET-DPO | $43.7 \pm 3.3$ | $47.0 \pm 1.0$ | $42.7 \pm 3.6$ | $85.5 \pm 8.5$ | $84.8 \pm 8.7$ | $80.9 \pm 9.4$ | $57.4 \pm 6.6$ |
| Diff | $45.1 \pm 3.0$ | $47.9 \pm 2.3$ | $52.8 \pm 2.9$ | $99.2 \pm 0.8$ | $98.4 \pm 1.1$ | $91.8 \pm 0.8$ | $59.0 \pm 1.1$ |
| Diff-CPL | $48.6 \pm 2.2$ | $45.9 \pm 2.8$ | $55.2 \pm 5.7$ | $\mathbf{100.0 \pm 0.0}$ | $99.6 \pm 0.2$ | $94.2 \pm 0.2$ | $63.5 \pm 0.8$ |
| FKPD | $51.2 \pm 0.7$ | $58.5 \pm 2.5$ | $64.4 \pm 2.7$ | $99.8 \pm 0.3$ | $98.3 \pm 1.4$ | $89.5 \pm 2.9$ | $64.1 \pm 3.2$ |
| Diff-UAPA-C | $\mathbf{56.1 \pm 0.9}$ | $\mathbf{61.3 \pm 2.2}$ | $\mathbf{68.1 \pm 0.6}$ | $\mathbf{100.0 \pm 0.0}$ | $99.7 \pm 0.2$ | $95.4 \pm 0.6$ | $70.9 \pm 2.5$ |
| Diff-UAPA-I | $54.3 \pm 1.1$ | $59.9 \pm 1.7$ | $66.2 \pm 1.3$ | $99.9 \pm 0.1$ | $\mathbf{99.8 \pm 0.2}$ | $\mathbf{95.7 \pm 1.9}$ | $\mathbf{71.7 \pm 4.6}$ |

(Rafailov et al., 2023) and contrastive preference learning (Hejna et al., 2024) to align the BET model. For diffusion-based policies, we introduce: 3) Diffusion Policy-CPL (**Diff-CPL**) that uses the MLE loss for aligning the diffusion policy (Obj. 12), and 4) **FKPD** (Shan et al., 2024) that performs forward KL regularized preference optimization. For our Diff-UAPA algorithm, we explore two distinct strategies for updating the Beta prior model: 5) **Diff-UAPA-C** that trains the Beta model using full preference data across the iterations without updates, and 6) **Diff-UAPA-I** that incrementally updates the Beta model on the current noisy preference data through the iterative process.

### 5.1. Model Performance in Robot Manipulation Tasks

**Task Description.** In this experiment, we evaluate the model's performance across three tasks from Robomimic (Mandlekar et al., 2021) and the Franka Kitchen task introduced in (Gupta et al., 2019), both of which use state-based observations. Specifically, the three Robomimic tasks—Lift, Can, and Square—address different manipulation challenges in a simulated environment, including object lifting, can manipulation, and square positioning. On the other hand, the Franka Kitchen task involves complex, multi-step, long-horizon activities that require interactions with seven distinct objects, with the objective to complete as many demonstrated tasks as possible, regardless of the execution order. Following Chi et al. (2023), we use *success rate* as the primary evaluation metric. For each task, the reference policy $\pi_{\text{ref}}$ is trained to achieve a success rate of approximately 40%. We then roll out the policy to collect 560 trajectories per task and construct the preference dataset based on their rewards. Please check Appendix D.2

for environmental details and Appendix D.3 for details on preference dataset construction.

**Results Analysis.** Table 1 presents the evaluation performance across three Robomimic tasks and the more complex Kitchen task. The results indicate that both variants of Diff-UAPA consistently outperform other methods across different tasks. This is primarily due to their use of a Beta prior, which effectively captures the uncertainty arising from potentially inconsistent preferences, thereby enhancing the diffusion policy training process. Moreover, the performance gap between Diff-UAPA-C and Diff-UAPA-I is relatively small, suggesting that the Beta prior can be trained effectively in both approaches, depending on the specific practice. This flexibility enhances the practical applicability of the proposed method. Notably, for the long-horizon Kitchen task, Diff-UAPA-I, which trains the Beta model incrementally, slightly outperforms Diff-UAPA-C, which pre-trains the Beta model using the complete dataset. This difference can be attributed to the fact that incremental training allows the model to adapt more dynamically to the changing preferences and environmental conditions over time, whereas pre-training may not fully capture such variability. We also provide the visualization results in Figure 2 in Appendix D.5

### 5.2. Model Performance in Locomotion Tasks

**Task Description.** The primary goal of Preference-based Reinforcement Learning (PbRL) is to align policies with *human* preferences. In this section, we assess the performance of Diff-UAPA using real human preferences provided by the Uni-RLHF benchmark (Yuan et al., 2024) in the HalfCheetah and Walker environments from the D4RL

*Table 2.* Episodic rewards of all methods in the HalfCheetah and Hopper environments with real human preferences.

| | BET | BET-CPL | BET-DPO | Diff | Diff-CPL | FKPD | Diff-UAPA-C | Diff-UAPA-I |
|---|---|---|---|---|---|---|---|---|
| HalfCheetah | $2577 \pm 198$ | $2976 \pm 66$ | $2948 \pm 37$ | $2838 \pm 325$ | $3121 \pm 148$ | $3060 \pm 201$ | $\mathbf{3399 \pm 72}$ | $3297 \pm 101$ |
| Hopper | $1161 \pm 90$ | $1226 \pm 85$ | $1129 \pm 79$ | $1296 \pm 137$ | $1313 \pm 103$ | $1370 \pm 120$ | $\mathbf{1591 \pm 51}$ | $1499 \pm 70$ |

benchmark (Fu et al., 2020). To ensure the dataset encompasses a diverse range of trajectories for meaningful comparison, we use *medium-expert* datasets for both environments. These datasets combine expert demonstrations from a near-optimal policy with suboptimal data generated by a medium-performing policy. Please check Appendix D.2 for more environmental details.

**Results Analysis.** The empirical results for the locomotion tasks are presented in Table 2. We observe that Diff-UAPA consistently outperforms other baselines across both environments. The key reason for this is that, during the iterative preference alignment process, some trajectory pairs may receive inconsistent preference labels. These noisy labels introduce greater uncertainty, making it challenging for the policy to accurately assess the true value of these trajectories and replicate the higher-performing ones. Diff-UAPA effectively addresses this challenge by leveraging a prior model that captures this uncertainty, enabling the policy to evaluate the trajectories more fairly and reliably, which in turn leads to improved overall performance. We also observe that diffusion-based policies generally achieve better results than Gaussian-based policies, primarily due to their superior modeling capabilities, which becomes more crucial when accounting for underlying uncertainties.

### 5.3. Experiments on Noise Sensitivity

**Task Description.** In this section, we perform a noise sensitivity evaluation in the Franka Kitchen environment to assess the robustness of different methods. Specifically, we adjust the reversal rate $r$ from 50% (as used in previous experiments) to 25% and 75%, to evaluate the method's stability under different levels of inconsistency. For clarity, we present only the most challenging p4 metric.

*Table 3.* Evaluation results of p4 metric under different levels of reverse rates in the Kitchen environment.

| | r=25% | r=50% | r=75% |
|---|---|---|---|
| BET-CPL | $65.7 \pm 1.6$ | $62.6 \pm 2.0$ | $55.0 \pm 2.5$ |
| BET-DPO | $60.2 \pm 4.8$ | $57.4 \pm 6.6$ | $47.2 \pm 7.0$ |
| Diff-CPL | $66.0 \pm 1.0$ | $63.5 \pm 0.8$ | $57.1 \pm 2.5$ |
| FKPD | $71.3 \pm 2.3$ | $64.1 \pm 3.2$ | $62.3 \pm 4.6$ |
| Diff-UAPA-C | $75.3 \pm 2.9$ | $70.9 \pm 2.5$ | $\mathbf{70.5 \pm 3.8}$ |
| Diff-UAPA-I | $\mathbf{75.5 \pm 3.0}$ | $\mathbf{71.7 \pm 4.6}$ | $69.1 \pm 5.2$ |

**Results Analysis.** Table 3 presents the evaluation results. As the noise level increases (i.e., the reversal rate), all methods show a decline in performance, highlighting the significance of uncertainties in the dataset. However, compared to the other methods, Diff-UAPA consistently exhibits better performance with the highest success rate regardless of the scale of noise. This underscores the effectiveness of incorporating the Beta prior model to handle such uncertainties.

## 6. Limitation

**Offline Trajectory Dataset.** This paper primarily focuses on learning from an offline trajectory dataset with potentially inconsistent human preferences that are iteratively updated, where the agent cannot directly interact with the environment. This partial offline setup may limit the agent's ability to explore and discover improved strategies through interactive online learning. However, our method can also generalize to an online setting, where both trajectories and human preferences are dynamically updated over time.

**Computational Overhead.** The integration of training a Beta prior model through variational inference adds computational complexity compared to simpler MLE-based methods. However, by utilizing efficient techniques like the reparameterization trick to enhance scalability, the computational overhead of training the Beta model is minimal in practice, adding only a small additional time cost relative to the diffusion training process.

## 7. Conclusion

In this paper, we present an uncertainty-aware preference alignment approach for diffusion policies using an iteratively updated preference dataset. Building on the maximum likelihood objective for directly aligning diffusion policies without learning a reward model, we introduce a Maximum A Posteriori (MAP) objective with an informative Beta prior, which is capable of capturing the uncertainty arising from potentially inconsistent human preferences. Empirical results across various domains demonstrate the effectiveness of our method. For future work, we extend this framework to the online RL setting with complex tasks involving humanoid robots or dexterous hand. By enabling agents to interact with the environment, our system can dynamically adapt to evolving human preferences, thereby solving more difficult applications.

## Impact Statement

The potential broader impact of this work is significant, as it advances the field of human-aligned decision-making in artificial intelligence (AI) and robotics. From an ethical perspective, this work emphasizes reducing bias and inconsistency in preference-based reinforcement learning, which aligns with the principles of fairness and equity in AI. However, challenges remain in ensuring the informed collection of preference data and safeguarding against misuse, such as exploiting preference alignment for manipulative or unethical purposes. Transparency in how user preferences are modeled and incorporated into decision-making policies is crucial to building trust and accountability.

Future societal consequences may include the development of AI systems that better reflect the diverse needs of global populations, contributing to more personalized and human-centric technologies. However, there is also the risk of over-reliance on population-level preferences that might inadvertently marginalize minority views or lead to unintended consequences if preferences are improperly interpreted or misaligned with ethical considerations. Addressing these risks requires careful oversight, interdisciplinary collaboration, and ongoing dialogue with diverse stakeholders.

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

# A. More Details in Section 4.1

We detailed the deviation from Equation (11) to Equation (12) here.

$$
\mathcal{L}_{1,\mathrm{MLE}}^{(\tau^w,\tau^l)}(\theta)
$$

$$
= -\log\sigma\Big(\alpha\cdot\Big(\sum_{t=0}^{k}\mathbb{E}_{\pi_\theta(a_t^{1:I,w}|s_t^w,a_t^{0,w})}\Big[\gamma^t\log\frac{\pi_\theta(\overline{a_t^w}|s_t^w)}{\pi_{\mathrm{ref}}(\overline{a_t^w}|s_t^w)}\Big] - \sum_{t=0}^{k}\mathbb{E}_{\pi_\theta(a_t^{1:I,l}|s_t^l,a_t^{0,l})}\Big[\gamma^t\log\frac{\pi_\theta(\overline{a_t^l}|s_t^l)}{\pi_{\mathrm{ref}}(\overline{a_t^l}|s_t^l)}\Big]\Big)\Big)
$$

$$
= -\log\sigma\Big(\alpha\cdot\Big(\sum_{t=0}^{k}\mathbb{E}_{\pi_\theta(a_t^{1:I,\cdot}|s_t^\cdot,a_t^{0,\cdot})}\Big[\gamma^t\log\frac{\pi_\theta(\overline{a_t^w}|s_t^w)}{\pi_{\mathrm{ref}}(\overline{a_t^w}|s_t^w)} - \gamma^t\log\frac{\pi_\theta(\overline{a_t^l}|s_t^l)}{\pi_{\mathrm{ref}}(\overline{a_t^l}|s_t^l)}\Big]\Big)\Big)
$$

$$
= -\log\sigma\Big(\alpha\cdot\Big(\sum_{t=0}^{k}\mathbb{E}_{\pi_\theta(a_t^{1:I,\cdot}|s_t^\cdot,a_t^{0,\cdot})}\Big[\sum_{i=1}^{I}\Big(\gamma^t\log\frac{\pi_\theta(a_t^{i-1|i,w}|s_t^w)}{\pi_{\mathrm{ref}}(a_t^{i-1|i,w}|s_t^w)} - \gamma^t\log\frac{\pi_\theta(a_t^{i-1|i,l}|s_t^l))}{\pi_{\mathrm{ref}}(a_t^{i-1|i,l}|s_t^l))}\Big)\Big]\Big)\Big)
$$

$$
= -\log\sigma\Big(\alpha\cdot\Big(\mathbb{E}_{\pi_\theta(a_\cdot^{1:I,\cdot}|s_t^\cdot,a_t^{0,\cdot})}\Big[\sum_{t=0}^{k}\sum_{i=1}^{I}\Big(\gamma^t\log\frac{\pi_\theta(a_t^{i-1|i,w}|s_t^w)}{\pi_{\mathrm{ref}}(a_t^{i-1|i,w}|s_t^w)} - \gamma^t\log\frac{\pi_\theta(a_t^{i-1|i,l}|s_t^l)}{\pi_{\mathrm{ref}}(a_t^{i-1|i,l}|s_t^l)}\Big)\Big]\Big)\Big)
$$

$$
= -\log\sigma\Big(\alpha I\cdot\Big(\mathbb{E}_{\substack{a_t^{i,w}\sim q(a_t^i|s_t^w,a_t^{0,w})\pi_\theta(a_t^{0,w}|s_t^w,a_t^{i,w}),\\ a_t^{i,l}\sim q(a_t^i|s_t^l,a_t^{0,l})\pi_\theta(a_t^{i-1,l}|s_t^l,a_t^{i,l})}}\Big[\sum_{t=0}^{k}\Big(\gamma^t\log\frac{\pi_\theta(a_t^{i-1|i,w}|s_t^w)}{\pi_{\mathrm{ref}}(a_t^{i-1|i,w}|s_t^w)} - \gamma^t\log\frac{\pi_\theta(a_t^{i-1|i,l}|s_t^l)}{\pi_{\mathrm{ref}}(a_t^{i-1|i,l}|s_t^l)}\Big)\Big]\Big)\Big)
$$

Since $-\log\sigma(x)$ is a convex function:

$$
(-\log\sigma(x))'' = (\sigma(x)-1)' = (\sigma(x)(1-\sigma(x))) \geq 0
$$

According to Jensen's inequality:

$$
\mathbb{E}_{\substack{a_t^{i,w}\sim q(a_t^{i,w}|a_t^{0,w},s_t^w),\\ a_t^{i,l}\sim q(a_t^{i,l}|a_t^{0,l},s_t^l)}}\Big[-\log\sigma\Big(\alpha I\cdot\Big(\mathbb{E}_{a_t^{i-1,\cdot}\sim\pi_\theta(a_t^{i-1,\cdot}|s_t^\cdot,a_t^{0,\cdot})}[\sum_{t=0}^{k}(\gamma^t\log\frac{\pi_\theta(a_t^{i-1|i,w}|s_t^w)}{\pi_{\mathrm{ref}}(a_t^{i-1|i,w}|s_t^w)} - \gamma^t\log\frac{\pi_\theta(a_t^{i-1|i,l}|s_t^l)}{\pi_{\mathrm{ref}}(a_t^{i-1|i,l}|s_t^l)})]\Big)\Big)\Big]
$$

$$
= \mathbb{E}_{\substack{a_t^{i,w}\sim q(a_t^{i,w}|a_t^{0,w},s_t^w),\\ a_t^{i,l}\sim q(a_t^{i,l}|a_t^{0,l},s_t^l)}}\Big[-\log\sigma\Big(\alpha I\cdot\sum_{t=0}^{k}\Big(\gamma^t\mathbb{D}_{\mathrm{KL}}\Big[\pi_\theta(a_t^{i-1|i,w}|s_t^w)\,||\,\pi_{\mathrm{ref}}(a_t^{i-1|i,w}|s_t^w)\Big] - \gamma^t\mathbb{D}_{\mathrm{KL}}\Big[\pi_\theta(a_t^{i-1|i,l}|s_t^l)\,||\,\pi_{\mathrm{ref}}(a_t^{i-1|i,l}|s_t^l)\Big]\Big)\Big)\Big]
$$

According to Formula (1), it can be further simplified as:

$$
-\mathbb{E}_{\substack{a_t^{i,w}\sim q(a_t^{i,w}|a_t^{0,w},s_t^w),\\ a_t^{i,l}\sim q(a_t^{i,l}|a_t^{0,l},s_t^l)}}\Big[\log\sigma\Big(-\alpha I\cdot\Big(\sum_{t=0}^{k}\gamma^t(\|\epsilon^w - \epsilon_\theta(a_t^{i,w},s_t^w,i)\|_2^2 - \|\epsilon^w - \epsilon_{\mathrm{ref}}(a_t^{n,w},s_t^w,i)\|_2^2)
$$

$$
-\sum_{t=0}^{k}\gamma^t(\|\epsilon^l - \epsilon_\theta(a_t^{i,l},s_t^l,i)\|_2^2 - \|\epsilon^l - \epsilon_{\mathrm{ref}}(a_t^{i,l},s_t^l,i)\|^2)\Big)\Big)\Big]
$$

where 1) $i\sim\mathcal{U}(0,I)$ is the diffusion timestep, 2) $a_t^{i,w/l}\sim q(a_t^{i,w/l}|a_t^{0,w/l},s^{w/l}$ denotes the action $a_t^{0,w/l}$ corrupted with noise $\epsilon^{w/l}$ after $i$ diffusion steps, and 3) $\epsilon_\theta^{w/l}$ is the noise predictor.

# B. More Details in Section 4.2

We detailed the deviation of Equation (16) here.

$$\mathcal{L}_{1,\text{prior}}^{(\tau^w,\tau^l)}(\theta)$$

$$= -\log\sigma\Big(\text{Beta}\Big(\alpha\cdot\Big(\sum_{t=0}^{k}\mathbb{E}_{\pi_\theta(a_t^{1:I,w}|s_t^w,a_t^{0,w})}\Big[\gamma^t\log\frac{\pi_\theta(\overline{a_t^w}|s_t^w)}{\pi_{\text{ref}}(\overline{a_t^w}|s_t^w)}\Big]$$

$$-\sum_{\tau\in\mathcal{D}_{\text{pref}},t=0}^{k}\mathbb{E}_{\pi_\theta(a_t^{1:I,\tau}|s_t^\tau,a_t^{0,\tau})}\Big[\gamma^t\log\frac{\pi_\theta(\overline{a_t^\tau}|s_t^\tau)}{\pi_{\text{ref}}(\overline{a_t^\tau}|s_t^\tau)}\Big]\Big);\alpha+1,\beta+1\Big)\Big)$$

$$= -\log\sigma\Big(\text{Beta}\Big(\alpha\cdot\Big(\mathbb{E}_{\pi_\theta(a_t^{1:I,\cdot}|s_t,a_t^{0,\cdot})}\Big[\sum_{t=0}^{k}\gamma^t\log\frac{\pi_\theta(\overline{a_t^w}|s_t^w)}{\pi_{\text{ref}}(\overline{a_t^w}|s_t^w)}-\sum_{\tau\in\mathcal{D}_{\text{pref}},t=0}^{k}\frac{\gamma^t}{|\mathcal{D}_{\text{pref}}|}\log\frac{\pi_\theta(\overline{a_t^\tau}|s_t^\tau)}{\pi_{\text{ref}}(\overline{a_t^\tau}|s_t^\tau)}\Big]\Big);\alpha+1,\beta+1\Big)\Big)$$

$$= -\log\sigma\Big(\alpha\cdot\Big(\sum_{\tau\in\mathcal{D}_{\text{pref}},t=0}^{k}\mathbb{E}_{\pi_\theta(a_t^{1:I,\cdot}|s_t,a_t^{0,\cdot})}\Big[\sum_{i=1}^{I}\Big(\sum_{t=0}^{k}\gamma^t\log\frac{\pi_\theta(a_t^{i-1|i,w}|s_t^w)}{\pi_{\text{ref}}(a_t^{i-1|i,w}|s_t^w)}-\sum_{\tau\in\mathcal{D}_{\text{pref}},t=0}^{k}\frac{\gamma^t}{|\mathcal{D}_{\text{pref}}|}\log\frac{\pi_\theta(a_t^{i-1|i,\tau}|s_t^\tau))}{\pi_{\text{ref}}(a_t^{i-1|i,\tau}|s_t^\tau))}\Big)\Big]\Big)\Big)$$

$$= -\log\sigma\Big(\alpha\cdot\Big(\mathbb{E}_{\pi_\theta(a^{1:I,\cdot}|s_t,a_t^{0,\cdot})}\Big[\Big(\sum_{t=0}^{k}\sum_{i=1}^{I}\gamma^t\log\frac{\pi_\theta(a_t^{i-1|i,w}|s_t^w)}{\pi_{\text{ref}}(a_t^{i-1|i,w}|s_t^w)}-\sum_{\tau\in\mathcal{D}_{\text{pref}},t=0}^{k}\sum_{i=1}^{I}\frac{\gamma^t}{|\mathcal{D}_{\text{pref}}|}\log\frac{\pi_\theta(a_t^{i-1|i,\tau}|s_t^\tau)}{\pi_{\text{ref}}(a_t^{i-1|i,\tau}|s_t^\tau)}\Big)\Big]\Big)\Big)$$

$$= -\log\sigma\Big(\alpha I\cdot\Big(\mathbb{E}_{a_t^{i,\cdot}\sim q(a_t^i|s_t,a_t^{0,\cdot})\pi_\theta(a_t^{0,\cdot}|s_t,a_t^{i,\cdot})}\Big[\Big(\sum_{t=0}^{k}\gamma^t\log\frac{\pi_\theta(a_t^{i-1|i,w}|s_t^w)}{\pi_{\text{ref}}(a_t^{i-1|i,w}|s_t^w)}-\sum_{\tau\in\mathcal{D}_{\text{pref}},t=0}^{k}\frac{\gamma^t}{|\mathcal{D}_{\text{pref}}|}\log\frac{\pi_\theta(a_t^{i-1|i,\tau}|s_t^\tau)}{\pi_{\text{ref}}(a_t^{i-1|i,\tau}|s_t^\tau)}\Big)\Big]\Big)\Big)$$

Since $-\log\sigma(\text{Beta}(x;\alpha,\beta))$ is a convex function when $\alpha+\beta\geq 2$. Define $g(t)=-\log(\sigma(t))$. Since

$$-\log(\sigma(t))=\log(1+e^{-t}),$$

it suffices to show that $\log(1+e^{-t})$ is convex in $t$. Differentiating,

$$\frac{d}{dt}\log(1+e^{-t})=\frac{-e^{-t}}{1+e^{-t}}=-\frac{1}{e^t+1},$$

and hence

$$\frac{d^2}{dt^2}\log(1+e^{-t})=\frac{e^t}{(e^t+1)^2}>0\quad(\forall t\in\mathbb{R}).$$

This shows $\log(1+e^{-t})$ is strictly convex in $t$. Therefore, for the function

$$f(x)=-\log\Big[\sigma\big(\text{Beta}(x;\alpha+1,\beta+1)\big)\Big],$$

the inner part $\text{Beta}(x;\alpha+1,\beta+1)$ serves as the real argument $t$, and the composition preserves convexity, implying $f(x)$ is convex.

According to Jensen's inequality

$$\mathbb{E}_{\substack{\tau\in\mathcal{D}_{\text{pref}},\\a_t^{i,\cdot}\sim q(a_t^{i,\cdot}|a_t^{0,\cdot},s_t)}}\Big[-\log\sigma\Big(\text{Beta}\Big(\alpha I\cdot\Big(\mathbb{E}_{a_t^{i-1,\cdot}\sim\pi_\theta(a_t^{i-1,\cdot}|s_t,a_t^{0,\cdot})}[(\sum_{t=0}^{k}\gamma^t\log\frac{\pi_\theta(a_t^{i-1|i,w}|s_t^w)}{\pi_{\text{ref}}(a_t^{i-1|i,w}|s_t^w)}-\sum_{\tau\in\mathcal{D}_{\text{pref}},t=0}^{k}\frac{\gamma^t}{|\mathcal{D}_{\text{pref}}|}\log\frac{\pi_\theta(a_t^{i-1|i,\tau}|s_t^\tau)}{\pi_{\text{ref}}(a_t^{i-1|i,\tau}|s_t^\tau)})]\Big);\alpha+1,\beta+1\Big)\Big)\Big]$$

$$= \mathbb{E}_{\substack{\tau\in\mathcal{D}_{\text{pref}},\\a_t^{i,\cdot}\sim q(a_t^{i,\cdot}|a_t^{0,\cdot},s_t)}}\Big[-\log\sigma\Big(\alpha I\cdot\sum_{t=0}^{k}\Big(\gamma^t\mathbb{D}_{\text{KL}}\Big[\pi_\theta(a_t^{i-1|i,w}|s_t^w)\,||\,\pi_{\text{ref}}(a_t^{i-1|i,w}|s_t^w)\Big]-\sum_{\tau\in\mathcal{D}_{\text{pref}},t=0}^{k}\frac{\gamma^t}{|\mathcal{D}_{\text{pref}}|}\mathbb{D}_{\text{KL}}\Big[\pi_\theta(a_t^{i-1|i,\tau}|s_t^\tau)\,||\,\pi_{\text{ref}}(a_t^{i-1|i,\tau}|s_t^\tau)\Big]\Big)\Big)\Big]$$

According to Formula (1), it can be further simplified as:

$$-\mathbb{E}_{\substack{\tau \in \mathcal{D}_{\text{pref}}, \\ a_t^{i,\cdot} \sim q(a_t^{i,\cdot}|a_t^{0,\cdot}, s_t)}} \left[ \log \sigma\left( -\alpha I \cdot \left( \sum_{t=0}^{k} \gamma^t (\|\epsilon^w - \epsilon_\theta(a_t^{i,w}, s_t^w, i)\|_2^2 - \|\epsilon^w - \epsilon_{\text{ref}}(a_t^{n,w}, s_t^w, i)\|_2^2) \right.\right.\right.$$

$$\left.\left.\left. - \sum_{\tau \in \mathcal{D}_{\text{pref}}, t=0}^{k} \frac{\gamma^t}{|\mathcal{D}_{\text{pref}}|} (\|\epsilon^\tau - \epsilon_\theta(a_t^{i,\tau}, s_t^\tau, i)\|_2^2 - \|\epsilon^\tau - \epsilon_{\text{ref}}(a_t^{i,\tau}, s_t^\tau, i)\|^2) \right) \right) \right]$$

where 1) $i \sim \mathcal{U}(0, I)$ is the diffusion timestep, 2) $a_t^{i,\cdot} \sim q(a_t^{i,\cdot}|a_t^{0,\cdot}, s_\cdot)$ denotes the action $a_t^{0,\cdot}$ corrupted with noise $\epsilon^\cdot$ after $i$ diffusion steps, and 3) $\epsilon_\theta$ is the noise predictor.

## C. Proof of Proposition 4.1

Proposition 4.1 can be divided into two parts: 1) the uncertainty-aware property of the Beta prior, and 2) the prior on the strength of a trajectory.

**Part 1.** We show the uncertainty-aware capability of the Beta prior $\text{Beta}(\phi(\tau); \alpha, \beta)$ during the iterative preference alignment process outlined in Definition 3.1 as follows.

The probability density function (PDF) of the Beta distribution $\text{Beta}(\phi(\tau); \alpha, \beta)$ is given by:

$$f(\phi(\tau); \alpha, \beta) = \frac{\phi(\tau)^{\alpha-1}(1 - \phi(\tau))^{\beta-1}}{B(\alpha, \beta)}, \quad 0 \leq \phi(\tau) \leq 1, \tag{19}$$

where $B(\alpha, \beta) = \int_0^1 t^{\alpha-1}(1-t)^{\beta-1}\, dt$ is the Beta function, serving as a normalizing constant.

The variance of a Beta distribution $\text{Beta}(\phi(\tau); \alpha, \beta)$ is given by the following formula:

$$\text{Var}(\text{Beta}(\alpha, \beta)) = \frac{\alpha\beta}{(\alpha + \beta)^2(\alpha + \beta + 1)}. \tag{20}$$

In the process described in Definition 3.1, the uncertainty arises from the varying preferences of different human raters for a given trajectory pair $(\tau^i, \tau^j)$. Without loss of generality, assuming an initial belief of $\text{Beta}(1, 1)$ for each trajectory, and with 10 raters evaluating a candidate pair $(\tau^i, \tau^j)$, the Beta prior is updated according to the preferences expressed by the raters. For instance, in the first case, where 9 raters prefer $\tau^i$ and 1 rater prefers $\tau^j$, the Beta prior for $\tau^i$ would be updated to $\text{Beta}(10, 2)$. In the second case, where 5 raters prefer $\tau^i$ and 5 prefer $\tau^j$, the Beta prior for $\tau^i$ would become $\text{Beta}(6, 6)$. Intuitively, we would be more confident with less uncertainty in the first case, as the majority of raters share the same preference.

The Beta distribution effectively captures this uncertainty. As shown in Equation (20), the variance of $\text{Beta}(10, 2)$ is smaller than that $\text{Beta}(6, 6)$, indicating that $\text{Beta}(10, 2)$ is 'sharper' and reflects less uncertainty, which aligns with our intuition.

**Part 2.** We prove that the prior on the strength of a trajectory is proportional to $\text{Beta}((\phi(\tau); \alpha+1, \beta+1))$, i.e., $p_0(A^{\pi_\theta}(\tau)) \propto \text{Beta}(\phi(\tau); \alpha + 1, \beta + 1)$, as follows.

Recall that the probability of a trajectory $\tau$ with strength $A^{\pi_\theta}(\tau)$ winning against the average candidate is given by $\phi(\tau) = \sigma(A^{\pi_\theta}(\tau) - \bar{A}^{\pi_\theta}) \in (0, 1)$. Let $A^{\pi_\theta}(\tau) - \bar{A}^{\pi_\theta}$ be denoted as $\tilde{A}^{\pi_\theta}(\tau)$. According to Equation (19), we have that the Beta distribution over $\phi(\tau) = \sigma(\tilde{A}^{\pi_\theta}(\tau))$ is:

$$\text{Beta}(\sigma(\tilde{A}^{\pi_\theta}(\tau)); \alpha, \beta) \propto \sigma(\tilde{A}^{\pi_\theta}(\tau))^{\alpha-1}(1 - \sigma(\tilde{A}^{\pi_\theta}(\tau)))^{\beta-1}. \tag{21}$$

The derivative of the sigmoid function is:

$$\sigma'(\tilde{A}^{\pi_\theta}(\tau)) = \sigma(\tilde{A}^{\pi_\theta}(\tau))(1 - \sigma(\tilde{A}^{\pi_\theta}(\tau))). \tag{22}$$

By incorporating Equation (21) and Equation (22) into Equation (15), we have that:

$$p_0(A^{\pi_\theta}(\tau)) \propto \sigma(\tilde{A}^{\pi_\theta}(\tau))^\alpha (1 - \sigma(\tilde{A}^{\pi_\theta}(\tau)))^\beta$$
$$\propto \text{Beta}(\sigma(\tilde{A}^{\pi_\theta}(\tau)); \alpha + 1, \beta + 1)$$
$$= \text{Beta}(\phi(\tau); \alpha + 1, \beta + 1). \tag{23}$$

## D. More Experimental Details

### D.1. Experimental Settings

In this paper, we utilized a total of 4 NVIDIA GeForce RTX 3090 GPUs, each with 24 GB of memory. The random seeds used for the experiments were 42, 43, and 44. We trained the agents offline and selected the final epoch for evaluation across 56 parallel environments, each with 10 episodes. Additionally, we employed a transformer-based architecture for the Beta model as in the preference transformer (Kim et al., 2023).

### D.2. Environmental Details

**Manipulation Tasks.** Robomimic (Mandlekar et al., 2021) is a large-scale robotic manipulation benchmark designed to explore imitation learning and offline reinforcement learning (RL). It consists of five tasks, each with a proficient human (PH) teleoperated demonstration dataset, and four tasks also feature mixed proficient/non-proficient human (MH) demonstration datasets, resulting in a total of nine variants. In this paper, we focus on three tasks: Lift, Can, and Square. Specifically:

- Lift: The robot arm must lift a small cube. This is the simplest task.

- Can: The robot must move a Coke can from a large bin to a smaller target bin. This task is slightly more challenging than Lift, as picking up the can is more difficult than picking up the cube, and the can must be placed accurately in the target bin.

- Square: The robot is required to pick up a square nut and place it onto a rod. This task is significantly more difficult than Lift and Can, as it demands high precision to pick up the nut and insert it into the rod.

The Franka Kitchen is also a widely used environment for evaluating the performance of methods in learning complex, long-horizon tasks. Introduced in Relay Policy Learning (Gupta et al., 2019), the environment features seven objects for interaction and includes a human demonstration dataset consisting of 566 demonstrations, each completing four tasks in random order. The objective is to execute as many of the demonstrated tasks as possible, regardless of their order, highlighting both short-horizon and long-horizon multimodal capabilities.

**Locomotion Tasks.** We evaluate our locomotion tasks using the D4RL benchmark (Fu et al., 2020), which is widely used in reinforcement learning (RL) for continuous control tasks. In this paper, we focus on the Hopper and HalfCheetah environments. In these environments, the goal is to maximize the cumulative reward within a single episode by navigating a sequence of actions that optimize the agent's movement and efficiency. More specifically:

- Hopper: In this task, the agent controls a 2D hopping robot, with the objective of balancing and moving the robot forward using as few steps as possible.

- HalfCheetah: In this task, the agent controls a 2D robotic cheetah, aiming to run as fast as possible while maximizing speed and maintaining stability.

### D.3. Manipulation Preference Dataset

For the robot manipulation tasks, we train two policies using behavior cloning: the BET policy and the diffusion policy. Training proceeds until a 40% success rate is reached. To build the simulation environment, we deploy 56 parallel environments, each initialized with a different seed to ensure varied initial positions for the agent. We then collect 560 trajectories per policy. From these, we randomly select 500 trajectory pairs and label them based on the sum of their rewards. During training, each trajectory is sliced using the observed steps as the stride, and these segments are compared. In the iterative update process, for each update round, we randomly select 20% of the trajectory pairs and apply a 50% reversal rate by swapping the winner and loser. To improve stability and convergence, the learning rate is reset at the start of each round.

### D.4. Hyperparameters

Our experiments are primarily based on the codebase from (Chi et al., 2023). Therefore, we retain the same hyperparameters for training the diffusion policy as specified in (Chi et al., 2023) for each experiment. The specific hyperparameters for Diff-UAPA are listed in Table 4.

*Table 4.* List of the specific hyperparameters for the proposed Diff-UAPA. To ensure fair comparisons, we maintain consistency in other parameters of the same neural networks across different models.

| Parameters | Robomimic | Kitchen | D4RL |
|---|---|---|---|
| **General** | | | |
| Training Epochs | 600 | 600 | 600 |
| Episode Length | 400 | 280 | 1000 |
| **Beta Model** | | | |
| Network | 256 | 256 | 256 |
| Learning Rate | 2e-5 | 2e-5 | 3e-5 |
| Number of Attention Heads | 4 | 4 | 4 |
| Number of Layers | 2 | 2 | 1 |
| Batch Size | 32 | 32 | 64 |
| Initial Belief | $\alpha = \beta = 1$ | $\alpha = \beta = 1$ | $\alpha = \beta = 1$ |

### D.5. Visualization Results

Figure 2 presents visualization results from the manipulation tasks. It is evident that the baseline method, Diff-CPL, which is trained using the MLE objective, struggles to handle certain critical scenarios, particularly those involving noisy preferences.

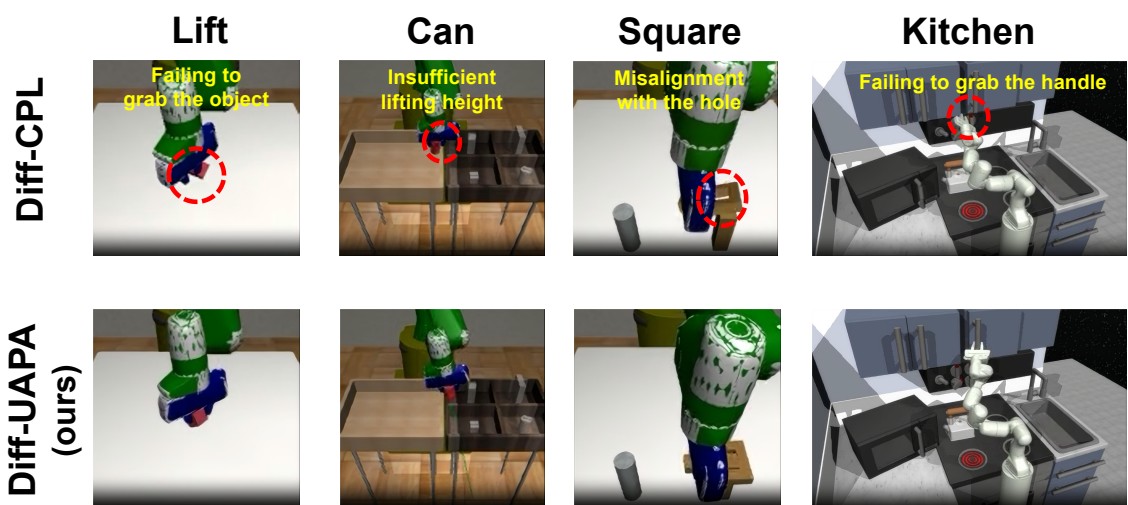

*Figure 2.* Visualization results in four manipulation tasks.

