# OpenReview forum: "Uncertainty-aware Preference Alignment for Diffusion Policies"
_ICML.cc/2025/Conference — Submitted to ICML 2025_

### Official Review · Reviewer_here · 2025-03-09

**Overall Recommendation:** 3

**Summary:**

This paper proposes Diff-UAPA, focusing on handling inconsistent and diverse offline preference data across different user groups.
Building upon diffusion policies, the authors first propose a maximum likelihood estimation (MLE) setup or preference alignment and then augment it with the Beta prior to capture the uncertainty, which is learned through variational inference.
Empirical results on some robotic manipulation tasks and D4RL tasks demonstrate the improved performance and stability of the proposed method under noisy feedback.

**Claims And Evidence:**

The claims are clean and the paper provides results to show improved performance and robustness in general.
However,  the evidence could be more convincing if the authors conduct more ablation studies to illustrate the contribution of the modeling of beta distribution or expand the range of benchmarks (e.g. medium-replay data in D4RL benchmarks).

**Essential References Not Discussed:**

N/A

**Experimental Designs Or Analyses:**

Regarding the experiments, although they demonstrate the method’s superior performance, they are somewhat limited in scope. More extensive ablation or sensitivity analyses would reinforce the paper’s claims. For instance, visualizing trajectories (e.g., in Maze2d) could more intuitively demonstrate the approach’s strengths.

**Methods And Evaluation Criteria:**

The proposed method aligns well with the problem. However, as the main components are each taken from other established works [1] and [2], it would be helpful if the authors clarified how their approach differs from or improves upon these earlier works.

[1] Wallace, B., Dang, M., Rafailov, R., Zhou, L., Lou, A., Purushwalkam, S., ... & Naik, N. (2024). Diffusion model alignment using direct preference optimization. In Proceedings of the IEEE/CVF Conference on Computer Vision and Pattern Recognition (pp. 8228-8238).

[2] Xu, S., Yue, B., Zha, H., and Liu, G. A distributional approach to uncertainty-aware preference alignment using offline demonstrations. In International Conference on Learning Representations, 2025.

**Other Comments Or Suggestions:**

Typos:
1. Lack of space in Line 66: (Diff-UAPA),a
2. Misuse of \citep and \citet in Line 173.
3. Wrong superscript in Line 174.
4. Abuse of notations: Should it be T instead of k in Equations 5, 6, 11, 12 ...

**Other Strengths And Weaknesses:**

Strengths:
1. It provides an iterative training procedure that can naturally adapt to changing user groups or inconsistent labeling in offline PbRL, which is beneficial for real-world applications.
2. The method creatively merges two lines of work based on max entropy RL.
3. The method shows demonstrable performance gains on several tasks

Weaknesses:
1. The key methodological components appear to be largely drawn from existing works, **without much additional design**. For example, the text from lines 220–257 on page 5 is very similar to Section 4 in [1].
2. While the experimental results cover several tasks, the breadth of testing could be expanded, perhaps with additional experiments or deeper analyses in the supplementary material.
3. There are a few typos (see other comments).

[1] Wallace, B., Dang, M., Rafailov, R., Zhou, L., Lou, A., Purushwalkam, S., Ermon, S., Xiong, C., Joty, S., and Naik, N. Diffusion model alignment using direct preference optimization. In Proceedings of the IEEE/CVF Conference on Computer Vision and Pattern Recognition, pp. 8228–8238, 2024

**Questions For Authors:**

Could you clarify how the two main methodological components (MLE-based preference alignment and  the Beta prior) differ in this work compared to [1]



[1] Wallace, B., Dang, M., Rafailov, R., Zhou, L., Lou, A., Purushwalkam, S., Ermon, S., Xiong, C., Joty, S., and Naik, N. Diffusion model alignment using direct preference optimization. In Proceedings of the IEEE/CVF Conference on Computer Vision and Pattern Recognition, pp. 8228–8238, 2024

[2] Xu, S., Yue, B., Zha, H., and Liu, G. A distributional approach to uncertainty-aware preference alignment using offline demonstrations. In International Conference on Learning Representations, 2025.

**Relation To Broader Scientific Literature:**

N/A

**Theoretical Claims:**

I have roughly checked the correctness of the proofs and did not find any obvious errors. However, the main theoretical claims seem largely derived from [1], weakening the contributions.

[1] Wallace, B., Dang, M., Rafailov, R., Zhou, L., Lou, A., Purushwalkam, S., ... & Naik, N. (2024). Diffusion model alignment using direct preference optimization. In Proceedings of the IEEE/CVF Conference on Computer Vision and Pattern Recognition (pp. 8228-8238).

---

> ### Author Rebuttal · Authors · 2025-04-01
>
> Dear Reviewer, we sincerely value your time and effort in evaluating our work. We have prepared comprehensive responses and clarifications to address each point you raised. We hope these responses can resolve your concerns.
>
> > Q1. The authors could conduct more ablation studies or expand the range of benchmarks (e.g. medium-replay data in D4RL benchmarks).
>
> **A1.** Thanks for your valuable suggestions. we conducted additional experiments, including more tasks and ablation studies. Please refer to E.1 and E.3 at https://anonymous.4open.science/r/Diff-UAPA-Rebuttal.
>
> ---
>
> > Q2. Could you clarify how the two main methodological components (MLE-based preference alignment and the Beta prior) differ in this work compared to [1][2].
>
> **A2.** Thank you for raising the question. We would like to provide a more detailed discussion highlighting the differences between this paper and the two prior works as follows.
>
> 1. **Difference between [1]**. We would like to highlight that the alignment for diffusion policy via MLE is not our primary contribution. Instead, we primally follow the approach developed for LLM in [1], and adapt it to the RL setting in our work. While we adopt some techniques from [1] (we have correctly cited [1]), the problem setting, preference model, and derivation are different. Specifically,
>
>     - **Problem setting.** [1] is formulated in the context of **LLM alignment**, where rewards are assigned exclusively at the final step, and preferences are based solely on the **final output** of the LLM. This approach optimizes the LLM’s ultimate output (see Eq. 14 in [1]). In contrast, Eq. 12 in this paper extends the framework to a **trajectory-wise** setting within the **RL field**. The key distinction is that in RL, we incorporate intermediate rewards. As a result, this paper formulates preference alignment based on the entire trajectory rather than focusing solely on the final state-action pair.
>
>     - **Preference model and deviation.** [1] employs a **reward-based preference** model while regularizing the **KL-divergence** with respect to a reference policy (see Eq. 3 in [1]). In contrast, this work adopts a **regret (advantage)-based preference** model (see Eq. 5) within the **maximum entropy RL** framework. To achieve trajectory-wise alignment under the advantage-based preference model, we define the chain advantage function (i.e., Eq. 8) and compute its expected value with respect to the diffusion latent variable.
>
>
> 2. **Difference between [2]**. While both works utilize the Beta prior with a MAP objective to model uncertainty during the alignment process, this work differs significantly regarding the problem formulation, motivation, and approach to incorporating the Beta prior. Specifically,
>
>     - **Problem formulation and motivation**. [2] addresses **epistemic uncertainty** from an offline preference dataset with imbalanced comparison frequencies across trajectories, where fewer compared trajectories induce greater uncertainty in reward prediction. In contrast, our work targets **aleatoric uncertainty** in human preferences, which arises from the inconsistent preferences from different annotator groups for the same trajectory pair. In other words, by interpreting the parameters $\alpha$ and $\beta$ of the Beta distribution as counts of 'vote' and 'unvote' feedback, [1] models the difference in their absolute values across trajectories (e.g., Beta(10,2) vs. Beta(100,20), where the former shows **greater uncertainty due to fewer counts**). In contrast, this work uses the Beta prior to model the relative strength of $\alpha$ and $\beta$ for different $\tau$ (e.g., Beta(6,6) vs. Beta(10,2), where the former shows **greater uncertainty due to vote inconsistency**).
>
>     - **Approach.** [2] adopts a **two-step** procedure, proposing a MAP objective for learning a **distributional reward model**. To achieve this, [2] introduces an iterative update rule that refines the reward model using the learned Beta model, which is then used for policy learning. In contrast, this work derives a unified MAP objective for directly aligning the **diffusion policy** in a **single-step** process. By maximizing the likelihood of the diffusion policy's output under the learned Beta distribution, the process guides the policy to align the estimated $\phi(\tau)$ with their prior distribution $p_0(\phi(\tau))$, which is more straightforward and efficient.
>
> ---
>
> > Q3. It would be more user-friendly if they included a README with clear instructions on how to run the code.
>
> **A3.** We have included a README file to assist with running the code. Please check https://anonymous.4open.science/r/Diff-UAPA.
>
> ---
>
> > Q4. Some typos.
>
> **A4.** Thank you for your valuable suggestions. We have corrected them accordingly.
>
> ---
>
> **References**
>
> [1] Diffusion model alignment using direct preference optimization. CVPR 2024
>
> [2] A Distributional Approach to Uncertainty-Aware Preference Alignment Using Offline Demonstrations. ICLR 2025.

---

> > ### Comment · Reviewer_here · 2025-04-03
> >
> > Thanks for the explanation. I will keep my recommendation for now, but will keep watching the progress of other reviewers' interactions.

---

> > > ### Author Response · Authors · 2025-04-03
> > >
> > > Thanks for your acknowledgment, and we appreciate the time and effort you have taken to review our work. Your insightful feedback has been invaluable in refining our research.

---

### Official Review · Reviewer_hsYP · 2025-03-14

**Overall Recommendation:** 3

**Summary:**

This paper proposes Diff-UAPA, an uncertainty-aware preference alignment method for diffusion policy, designed to address inconsistencies in preference pairs. Diff-UAPA uses a maximum posterior objective to align the diffusion policy with a regret-based preference model, incorporating an informative Beta prior to mitigate these inconsistencies. Extensive experiments demonstrate the effectiveness of the proposed method.

## Update After Rebuttal: I raised my score from 2 to 3 as the authors' rebuttal addressed most of my concerns.

**Claims And Evidence:**

Yes, the design of the proposed learning framework has the potential to address inconsistencies and noisy preference labels in the dataset.

**Essential References Not Discussed:**

There are several papers addressing noisy preference labels, including approaches such as data filtering, label smoothing, and robust loss functions.

**Experimental Designs Or Analyses:**

There are many methods designed to be robust against noisy preferences, but the paper does not consider them. Including these methods would improve the soundness and validity of the experimental results.

**Methods And Evaluation Criteria:**

Yes, the methods and evaluation criteria are reasonable for real-world applicability.

**Other Comments Or Suggestions:**

Line 174 & 293: Use textual citations instead of parenthetical citations.

Line 175: The in-text math equation appears to be incorrect.

**Other Strengths And Weaknesses:**

Strengths

- The paper is well-written and easy to follow.
- The final performance of the proposed method appears strong.

Weaknesses

- The proposed method incorporates multiple existing components, such as using a diffusion policy instead of a simple MLP and a beta prior instead of a uniform prior. This complexity makes the algorithm difficult to interpret, particularly in understanding the contribution of each component. A more detailed ablation study with a decoupled analysis of each effect would help clarify the impact of these choices.

- The use of a diffusion policy and beta prior introduces computational overhead. A head-to-head comparison with other baselines would provide a clearer assessment of the method’s effectiveness.

- The paper lacks robust preference learning baselines (e.g., data filtering, label smoothing, and robust loss functions) and does not explore different types of noisy preference setups. Incorporating such setups, as discussed in [1], would strengthen the evaluation.

Reference:
[1] Robust Reinforcement Learning from Corrupted Human Feedback. Bukharin et al., NeurIPS 2024.

**Questions For Authors:**

Please refer to the "Other Strengths And Weaknesses" section above.

**Relation To Broader Scientific Literature:**

The PbRL framework, designed to address noisy and inconsistent preferences, is an important contribution to the broader scientific literature.

**Theoretical Claims:**

The derivation of each loss term appears sound and valid.

---

> ### Author Rebuttal · Authors · 2025-04-01
>
> Dear Reviewer, we sincerely value your time and effort in evaluating our work. We have prepared comprehensive responses and clarifications to address each point you raised. We hope these responses can resolve your concerns.
>
> > Q1. There are many methods designed to be robust against noisy preferences , including these robust preference learning baselines (e.g., data filtering, label smoothing, and robust loss functions) would improve the experiments. In addition, the paper does not explore different types of noisy preference setups.
>
> **A1.** We sincerely thank the reviewer for highlighting this valuable concern. In response to your suggestions, we have conducted **additional experiments across three distinct noisy preference setups** as described in [1], including stochastic noise, myopic noise, and irrational noise. We also incorporated a wider range of **baseline methods that are robust to noisy preferences**, such as $R^3M$ [1], RIME [2], and UA-PbRL [3]. For detailed results and discussions, please refer to Section E.4 at https://anonymous.4open.science/r/Diff-UAPA-Rebuttal.
>
> ---
>
> > Q2. There are several papers addressing noisy preference labels.
>
> **A2.** Thank you for raising this concern. We would like to emphasize that, as defined in Definition 3.1, this paper considers the iterative preference alignment setting, where the preference dataset is updated in each round (potentially with inconsistencies), thus requiring the learned policy to adapt to the new preferences progressively. As shown in Proposition 4.1, the proposed prior model could capture the uncertainty within the process. In contrast, prior works on robust PbRL typically assume a static preference dataset without updates, making them not directly work for this iterative setting.
>
> However, we acknowledge the importance of robustness and its tight relationship with uncertainty. In the revised paper, we have expanded the related works section to provide a more detailed discussion of existing studies on robustness in the context of noisy preference labels, including data filtering, label smoothing, and robust loss functions [1-7].
>
> ---
>
> > Q3. The proposed method incorporates multiple existing components, such as using a diffusion policy instead of a simple MLP and a beta prior instead of a uniform prior. A more detailed ablation study with a decoupled analysis of each effect would help clarify the impact of these choices.
>
> **A3.** Thank you for your valuable suggestions. We acknowledge the importance of a more detailed ablation study and analysis,  and have conducted **ablation studies on the two components** individually. For further details, please refer to Section E.3 at https://anonymous.4open.science/r/Diff-UAPA-Rebuttal.
>
> ---
>
> > Q4. The use of a diffusion policy and beta prior introduces computational overhead. A head-to-head comparison with other baselines would provide a clearer assessment of the method’s effectiveness.
>
> **A4.** Thank you for raising this concern. The additional computational overhead can primarily be attributed to the following components:
>
> - **Diffusion policy**. While diffusion policies incur higher computational costs than simpler architectures like MLPs, this overhead is partially offset by the action sequence prediction strategy in [8]. More importantly, diffusion models are widely adopted in RL for their strong generative capabilities and superior performance. In practice, training time for diffusion is roughly twice that of the transformer in our experiments.
>
> - **Beta model**. In this work, we use efficient techniques like the reparameterization trick to improve scalability. In practice, the computational cost of training the Beta model is **similar to training a reward model** in traditional PbRL. Since our method avoids training a reward model, the added cost is less effective compared to conventional PbRL. Additionally, the extra computational cost only slightly increases training time—by a few minutes—while the subsequent RL phase is much more demanding, often taking several hours.
>
> ---
>
> > Q5. Some typos.
>
> **A5.** Thank you for your valuable suggestions. We have corrected them accordingly and thoroughly checked the paper to avoid similar issues.
>
> ---
>
> **References**
>
> [1] Robust reinforcement learning from corrupted human feedback. NeurIPS 2024.
>
> [2] Rime: Robust preference-based reinforcement learning with noisy preferences. ICML 2024.
>
> [3] A Distributional Approach to Uncertainty-Aware Preference Alignment Using Offline Demonstrations. ICLR 2025.
>
> [4] Corruption robust offline reinforcement learning with human feedback. arXiv:2402.06734.
>
> [5] Sample selection with uncertainty of losses for learning with noisy labels. arXiv:2106.00445.
>
> [6] A note on dpo with noisy preferences \& relationship to ipo. 2023,
>
> [7] Distributionally Robust Reinforcement Learning with Human Feedback. arXiv:2503.00539.
>
> [8] Diffusion policy: Visuomotor policy learning via action diffusion. IJRR 2023.

---

> > ### Comment · Reviewer_hsYP · 2025-04-09
> >
> > Thank you to the authors for the detailed and thoughtful rebuttal. I gained a clearer understanding from the additional experiments. However, I still have some uncertainties regarding the advantages of the proposed method compared to a DPO framework combined with robustness formulations.
> >
> > First, the authors argue that the proposed method outperforms robust PbRL methods due to its iterative preference alignment design. However, I believe that robust PbRL methods can also be adapted to the iterative setting by explicitly or implicitly retraining their reward models across rounds.
> >
> > Second, as I understand it, the proposed method is effectively a two-stage approach: (1) learning the Beta prior and (2) updating the policy. If that is the case, I am curious about the benefit of this two-stage formulation over a unified one-stage approach such as DPO.
> >
> > Any clarification or additional insight on these points would help me better understand the effectiveness and distinct advantages of the proposed method.

---

> > > ### Author Response · Authors · 2025-04-09
> > >
> > > Dear Reviewer, we sincerely appreciate your constructive feedback and are grateful for the time and effort you've invested in reviewing our work. We hope the following response can address your remaining concerns in two points.
> > >
> > > > *"First, the authors argue that the proposed method outperforms robust PbRL methods due to its iterative preference alignment design. However, I believe that robust PbRL methods can also be adapted to the iterative setting by explicitly or implicitly retraining their reward models across rounds."*
> > >
> > > **Response.** Thanks for your question. We would like to provide a more detailed explanation of the distinct mechanisms between the robust PbRL methods and our method, particularly in how each addresses the "outlier" samples in the dataset.
> > >
> > > - Robust PbRL methods (e.g., data filter), generally aim to **exclude noisy or inconsistent data from the training process**. While this may help reduce the impact of outliers, it also risks discarding valuable information if certain data points are mistakenly deemed as outliers. This filtering approach can result in lost opportunities for learning from diverse, potentially useful preferences.
> > >
> > > - In contrast, our method employs an uncertainty-aware framework, which utilizes the Beta prior for handling uncertainties. **Rather than discarding uncertain data points, our approach assigns them lower confidence values**, which effectively down-weights their influence on the policy learning. This means that outliers or uncertain samples are not removed outright but are treated more conservatively. By modeling uncertainties, our method ensures these samples contribute to learning while minimizing their negative impact on the policy.
> > >
> > > In the iterative alignment process, some data points may shift in each round. By simply "retraining their reward models across rounds", the reward model is very likely to disregard the "outliers" within the single round. However, the proposed method captures these potential inconsistencies **throughout the entire learning process** (i.e., across rounds, updated iteratively), enhancing overall performance.
> > >
> > > ---
> > >
> > > > *"Second, as I understand it, the proposed method is effectively a two-stage approach: (1) learning the Beta prior and (2) updating the policy. If that is the case, I am curious about the benefit of this two-stage formulation over a unified one-stage approach such as DPO."*
> > >
> > > **Response.** Thank you for your question. We appreciate the opportunity to clarify the benefits of our two-stage approach.
> > >
> > > - The two-stage formulation in our method—(1) learning the Beta prior and (2) updating the policy—offers a clear advantage in terms of **uncertainty modeling**. The Beta prior in the first stage helps to explicitly **capture and account for the uncertainties arising from inconsistent human preferences**. This prior serves as **a guidance for subsequent policy updates**, allowing the method to effectively handle noisy or conflicting data without discarding valuable information.
> > >
> > > - In contrast, a one-stage approach such as DPO **directly optimizes the policy without explicitly modeling the underlying uncertainties**. While DPO can be effective in many cases, it may struggle when dealing with noisy, inconsistent, or evolving preferences.
> > >
> > > Our approach allows for a more principled treatment of these challenges by handling the uncertainty (via the Beta prior) in the first stage, which leads to more stable and reliable policy updates in the second stage. Thus, the two-stage approach used in this work provides a more structured and robust framework, especially in dynamic environments with varying levels of preference inconsistency. In addition, as discussed in the previous rebuttal, **the computational cost of training the Beta model is slight**. We believe this separation of concerns is what enables our method to achieve superior performance in practice.
> > >
> > > Thank you once again for your thoughtful question and the opportunity to elaborate further.

---

### Official Review · Reviewer_xmFi · 2025-03-16

**Overall Recommendation:** 3

**Summary:**

This paper proposes a method to align RL policy using human demonstration and preference feedback. The method works as follows: (1) learn a reference policy from a set of human demonstration trajectories via behavior cloning; (2) learn a prior distribution about the probability that a trajectory is preferred using a set of human preferences of pairs of trajectories; (3) align a discrete-time RL policy, parameterized by θ, where each action is running a diffusion policy for a fixed amount of time.

The key contribution is in step (2), where it learns a Beta prior distribution from the preference dataset that maps to one trajectory to the probability that it is preferred, and this distribution is represented by a transformer-based neural network. The motivation of this contribution is to allow the human preference dataset to be inconsistent, potentially due to the different populations who provide preference data.

Results are based on comparing the 2 implementations of the proposed method with 6 benchmark methods for simulated robotic tasks (3 tasks in Robomimic, 1 long horizon Franka Kitchen, 2 environments in D4RL) with real human preferences. The authors reverse some of the preference labels in the human preference dataset to simulate inconsistent preferences. The proposed method always achieved better performance. Also, as the number of inconsistent preference labels, performance drops, but the proposed method still performs the best relatively.

## update after rebuttal
I appreciate the authors' Rebuttal in addressing my concerns. I have adjusted my score accordingly.

**Claims And Evidence:**

* The main claim is that the proposed method can use inconsistent human preference data to learn RL policies where each action is one diffusion policy. This is supported by the empirical simulation study.
  * One limitation is that the proposed method also uses a reference policy, which is trained from some human demonstration data via behavior cloning. So, the proposed method uses both the demonstration data and the preference data. However, when I first read this paper, I didn't realize this until I saw Alg.1.
    * Since I didn't expect that the proposed method also uses reference policy, I got confused when reading Sec.3 and Sec.4. So it would be great to clarify the fact that the proposed method requires both human data throughout the paper from the introduction.
* A small claim made in the introduction is that one advantage of the proposed method is to "bypass the reward learning". This makes sense because the proposed method learns a prior distribution about the probability of a trajectory being preferred. However, it does not seem to be backed up by the empirical study. It might be useful to compare the performance of having a reward function vs not, or cite prior literature to show this.
* The paper is motivated by the inconsistency of human preferences. However, the empirical study seems to use datasets of consistent human preferences and convert them to inconsistent datasets. It would be stronger to motivate this work to choose some problem domains that human preferences datasets that are originally inconsistent.

**Essential References Not Discussed:**

No.

**Experimental Designs Or Analyses:**

Yes. The simulation study makes sense.

**Methods And Evaluation Criteria:**

Yes. The empirical study compares the proposed method with baselines to show the effectiveness of learning RL policies from demonstration and preference data.

**Other Comments Or Suggestions:**

NA

**Other Strengths And Weaknesses:**

## Strength
* The motivation is strong.
* The method is sound.

## Weakness
* One key weakness is that the paper is not very accessible, with several inconsistencies that make it hard for me to follow.
  * Eq.2 is very hard to understand. It has notations, such as ε_ref, that are not defined. I could not understand Eq.2 before understanding Eq.12.
    * Related to this, it seems that one loss function used in the proposed method, Eq.12, is a multi-step extension of Eq.2. If this is actually the key innovation behind Eq.12, then it would be great to clarify this early on when presenting Eq.2.
  * Sec.4.2 is also a bit hard to follow. I think it would be very helpful to explain what the prior p0(ϕ(τ)) and p0(A^πθ(τ)) means. When reading the paper, I got a bit confused about the notation p0(ϕ(τ)), because ϕ(τ) is a probability, and I guess p0(ϕ(τ)) is the density function about the probability ϕ(τ). It might be helpful to clarify that ϕ(τ) is the random variable, and p0(ϕ(τ))(.) is the density function.

* Eq.6 is obtained by plugging Eq.4 into Eq.5. However, this plugging-in operation relies on that the human's reference policy is actually π_ref. However, everyone who contributes to the preference dataset might have different reference policies, and these reference policies might be different from π_ref (learned via behavior cloning). It would be more convincing to motivate this assumption.
* Eq.13 is an assumption. It would be more convincing to discuss the implication or motivation of such an assumption.

**Questions For Authors:**

* Line 278 says the equation: P_MAP(A(τ)) ∝ p0(A(τ)) · P_MLE(A(τ)). I am a bit confused. Is it directly coming from Bayes rule?
* Sec.4.3 learns the prior p0(ϕ(τ) | D_pref). I wonder after you learn this prior as a transformer, how would you plug this learned function into Eq.16 to further optimize the policy?

**Relation To Broader Scientific Literature:**

This paper proposed a new method for RL from preference feedback, where each discrete RL action is a diffusion policy. It situates well in the field of using diffusion policy and RL for robotic applications.

**Theoretical Claims:**

I checked the proof of the key derivation for the proposed method (Prop.4.1), which is correct. I didn't check the derivation for Eq.12 and Eq.16.

---

> ### Author Rebuttal · Authors · 2025-04-01
>
> Dear Reviewer, we greatly appreciate your constructive comments. We have seriously considered your suggestions, and we hope the following response can address your concerns:
>
> > Q1. The proposed method uses both the demonstration data and the preference data.
>
> **A1.** Thank you for your comment. As shown in Definition 3.1 and Figure 1, the preference dataset is obtained by comparing samples from the trajectory dataset, so no additional demonstration dataset is required. Our pre-training is performed only with the trajectories in the preference dataset, following standard PbRL practices where pre-training is widely adopted (check "Pretraining" paragraph in [1]). To enhance clarity, we have updated the Introduction section to clarify this point.
>
> ---
>
> > Q2. One advantage of the proposed method is to "bypass the reward learning"...It is useful to compare the performance of having a reward function.
>
> **A2.** Thank you for raising this concern. The advantages of bypassing reward learning have been demonstrated in many studies, including robotics [1] and LLMs [2]. To better solve your concern, we conducted **additional experiments with two-step PbRL baselines**. Please refer to E.2 at https://anonymous.4open.science/r/Diff-UAPA-Rebuttal.
>
> ---
>
> > Q3. It would be stronger to motivate this work to choose some problem domains that human preferences datasets that are originally inconsistent.
>
> **A3.** Thank you for highlighting this. The experiments on D4RL (Section 5.2) utilize real human preferences from Uni-RLHF benchmark, which were collected from 100 annotators with diverse backgrounds. We believe these real human datasets are originally inconsistent.
>
> ---
>
> > Q4. It seems that Eq.12 is a multi-step extension of Eq.2...it would be great to clarify this early.
>
> **A4.** Sorry for the misunderstanding. $\epsilon_\theta$ represents the optimized diffusion policy parameters, and $\epsilon_{ref}$ represents the reference diffusion parameters. Regarding Eq.12, it can be viewed as a trajectory-wise extension of Eq.2 (step-wise) in the RL setting. To ensure clarity, we have updated the paper with more detailed descriptions when presenting Eq. 2.
>
> ---
>
> > Q5. Sec.4.2 is hard to follow...It might be helpful to clarify that $\phi(\tau)$ is the random variable, and $p_0(\phi(\tau))(\cdot)$ is the density function.
>
> **A5.** We apologize for any confusion. As defined in the sentences following Eq. 14 and Eq. 15, $\phi(\tau)\in(0,1)$ is the probability that a trajectory $\tau$ wins against the average candidate in the dataset, and it is a **Bernoulli random variable**. The prior $p_0(\phi(\tau))$ is indeed the **probability density function** of $\phi(\tau)$, which follows a Beta distribution, serving as the conjugate prior for the Bernoulli variable $\phi(\tau)$. It reflects our initial belief on the **winning probability** of different trajectories. Based on it, $p_0(A^{\pi_\theta}(\tau))$ defines our initial belief on the **strength** of different $\tau$. We have enhanced our presentation accordingly.
>
> ---
>
> > Q6. Eq.6 is obtained by plugging Eq.4 into Eq.5...However, everyone who contributes to the preference dataset might have different reference policies.
>
> **A6.** Thank you for raising the point. We would like to emphasize that, when learning from a single preference dataset, the reference policy acts as a third-party baseline, not necessarily aligned with personal preferences. It can be any fixed policy used for applying constraints or regularization during training. For example, in LLMs, the reference policy is usually obtained via Supervised Fine-Tuning, while in RL, it's often derived from BC.
>
> ---
>
> > Q7. Eq.13 is an assumption...discuss the motivation.
>
> **A7.** Thanks for your valuable advice. The assumption is based on the fact that preferences align with the negative discounted regret (i.e., $regret(\tau)=\sum_{t}\gamma^t regret(s_t,a_t)=-\sum_{t}\gamma^t A(s_t,a_t)$). Intuitively, we can define $A(\tau)=\sum_{t}\gamma^t A(s_t,a_t)$ to represent the negative regret $-regret(\tau)$. More details can be found in [1].
>
> ---
>
> > Q8. Line 278...Bayes rule?
>
> **A8.** Yes.
>
> ---
>
> > Q9. Sec.4.3 learns the prior $p_0(\phi(\tau)$...how to plug it into Eq.16 to further optimize the policy?
>
> **A9.** Thank you for your question. The prior model $p_0(\phi(\tau))$ encodes our learned belief about $\phi(\tau)$ for a given trajectory $\tau$. During policy optimization, for each input $\tau$, the diffusion policy computes $\phi(\tau)$ based on its definition, while the prior model outputs $[\alpha_\tau, \beta_\tau]$, which defines the target Beta distribution of $\phi(\tau)$. By maximizing the likelihood of the **diffusion model's output under this Beta distribution**, the process guides the policy optimization.
>
> ---
> **References**
>
> [1] Contrastive preference learning: Learning from human feedback without reinforcement learning. ICLR 2024.
>
> [2] Direct preference optimization: Your language model is secretly a reward model. NeurIPS 2023.

---

### Decision · Program_Chairs · 2025-05-01

**Decision:**

Reject

**Comment:**

This submission proposed Diff-UAPA, uncertainty-aware preference alignment for diffusion policies, to address the challenge of handling the inherent uncertainties in people's preferences during policy updates. While the motivation and the performance of the proposed method were recognized by the reviewers, some reviewers were concerned with the clarity, novelty, as well as insufficient experimental comparisons. The authors' rebuttal partially addressed those concerns, but there are still outstanding issues to be solved. Therefore, we cannot accept the current version of this submission to be published at ICML 2025.